# Crowding results from optimal integration of visual targets with contextual information

Guido Marco Cicchini [1], Giovanni D'Errico [1] & David Charles Burr [1,2] ✉

Crowding is the inability to recognize an object in clutter, usually considered a fundamental low-level bottleneck to object recognition. Here we advance and test an alternative idea, that crowding, like predictive phenomena such as serial dependence, results from optimizing strategies that exploit redundancies in natural scenes. This notion leads to several testable predictions: crowding should be greatest for unreliable targets and reliable flankers; crowding-induced biases should be maximal when target and flankers have similar orientations, falling off for differences around 20°; flanker interference should be associated with higher precision in orientation judgements, leading to lower overall error rate; effects should be maximal when the orientation of the target is near that of the average of the flankers, rather than to that of individual flankers. Each of these predictions were supported, and could be simulated with ideal-observer models that maximize performance. The results suggest that while crowding can affect object recognition, it may be better understood not as a processing bottleneck, but as a consequence of efficient exploitation of the spatial redundancies of the natural world.

Crowding is the inability to recognize and identify objects in clutter, despite their being clearly visible, and recognizable when presented in isolation[1] (see examples in Fig. 1a). It is particularly elevated in the periphery, impacts on many important daily tasks, such as face recognition and reading (for reviews see[2–4]), to the extent it has been considered a major bottleneck to object recognition.

There are several diagnostic criteria for crowding, the most important being that it scales linearly with eccentricity, such that the minimal spacing between centres of targets and flanking elements supporting uncrowded vision is equal to roughly half the target eccentricity (Bouma's law[5]). Another is that flankers similar (in colour, shape or orientation) to the target crowd more effectively than dissimilar ones[6–9]. Crowding is stronger in the upper than the lower visual field[10], and for radial than for tangential flankers[11].

Most popular current models of crowding involve some form of compulsory pooling (or substitution) of targets with flankers. For example, Parkes and colleagues[12] showed that while the orientation of a Gabor patch cannot be determined when embedded in flankers, it does influence the perceived orientation of the ensemble: hence it is merged with the flankers, rather than suppressed. This is reinforced by several studies showing that the targets can take on characteristics of the flanker stimuli[13–15]. The compulsory integration could occur in higher cortical areas, such as V4[2,16,17] or V2[18,19], which have large receptive fields, appropriately sized to account for Bouma's law.

However, compulsory integration is vague and does not explain all the known facts about crowding. For example, flankers that are similar in size, colour or orientation cause more crowding than dissimilar ones[9,20,21]. More difficult to explain are the recent demonstrations of Herzog and colleagues[22] of "uncrowding", where the addition of extra flanking stimuli around the flankers can reduce drastically their crowding effect, particularly if the extra flankers group with the original flankers to form coherent objects. These data do not fit easily with compulsory integration, even with appropriate linear filtering, which could in principle account for other effects, such as orientation or size selectivity.

Crowding has been studied for decades, and usually considered to be a defect in the system, "an essential bottleneck to object perception"[23]. Certainly, it impacts heavily on object recognition in

[1]Institute of Neuroscience, CNR, via Moruzzi, 1, 56124 Pisa, Italy. [2]Department of Neurosciences, Psychology, Drug Research and Child Health, University of Florence, viale Pieraccini, 6, 50139 Firenze, Italy. ✉e-mail: davidcharles.burr@unifi.it

a)

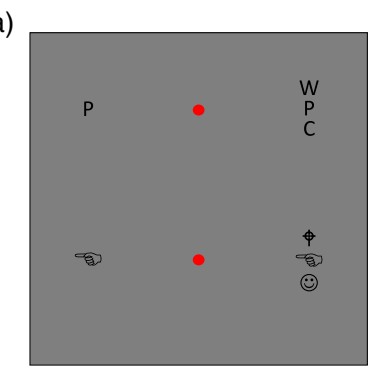

b)

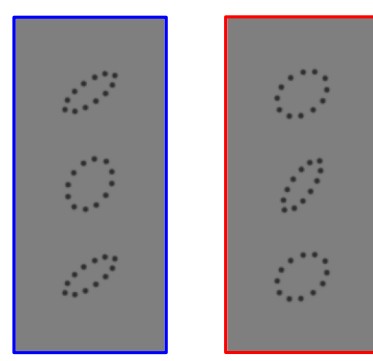

**Fig. 1 | Demonstration of visual crowding and examples of stimuli of this study. a** Crowding is a visual phenomenon where items that can be easily identified in isolation are not identifiable if surrounded by similar items. The P and hand symbol on the right are difficult to recognize while fixating the central red dots. **b** Stimuli employed in this experiment. Observers judged the orientation of a peripheral target (the central oval), which was flanked above and below by oval stimuli. Two conditions were tested: a rounded target with elongated flankers (Low reliability target, high reliability flankers, blue at left) or an elongated target with rounded ovals (red at right). In the main condition the centre-to-centre distance of flankers and targets was 5.5 deg, and eccentricity 26 degrees, leading to a Bouma ratio of 0.21.

tasks like letter or face recognition: but is it possible that it may reflect processes that are in principle advantageous to perception? Perception is strongly affected by contextual information, particularly temporal context, where recent and longer term perceptual history has been shown to exert a major influence on current perception[24–27]. While the role of context and experience has been appreciated for some time[28,29], it has become particularly topical in recent years within the framework of Bayesian analysis. This approach has revealed an interesting phenomenon termed "serial dependence", where the appearance of many important attributes of a stimulus (including orientation, numerosity, facial identity, beauty etc) are biased towards previously viewed stimuli[25,26,30,31]. Counterintuitively, these consistent biases in perception have been shown to reflect an efficient perceptual strategy, exploiting temporal redundancies in natural viewing to reduce overall reproduction errors, despite the biases[26,32,33].

Could crowding also be a consequence of efficient integration processes that exploit spatial (rather than temporal) redundancies to improve performance? We investigate this possibility by studying crowding with a paradigm similar to one used in serial dependence studies. If, like serial dependence, crowding is a by-product of efficient redundancy-reducing mechanisms, it should display several specific signature characteristics. One is that crowding-induced biases should be stronger for targets that are unreliably perceived, and for flankers that are reliably perceived. In addition, crowding should follow the signature pattern seen in serial dependence, highest when the orientations of target and flankers are similar, then steadily falling off. We provide evidence for these characteristics qualitatively and quantitatively, and show that crowding, while leading to biases, also improves overall performance. The results fit well with models simulating intelligent combination of signals from a small receptive field centred on the target with signals from a much larger integration region, following the same rules that govern serial dependence. On this view crowding is not merely a defect, or bottleneck, in the system, but the unavoidable consequence of efficient exploitation of spatial redundancies of the natural world.

## Results

To test if visual crowding follows the rules of optimal integration, which well describe serial dependence[25,33], we measured crowding with an orientation reproduction task. Participants reproduced the orientation of oval stimuli, which were either elongated (aspect ratio 1: 2.8) or rounded (1: 1.4). Targets were presented 26° to the right of fixation, and vertically flanked by similar oval stimuli, elongated if the target was rounded, and vice versa (see Fig. 1b). The orientation of the target was either 35° or 55° (at random). The orientations of the two flankers were yoked together, and varied randomly over a range of ± 45° from target orientation. The clear prediction from models of efficient integration[32,34] (see Eq. 10 & 15) is that the effects of crowding will be stronger for the unreliable targets and reliable flankers than vice versa, which we test with rounded targets and elongated flankers. The reasons are explained formally in the modelling section, but the intuition is that the rounded stimuli have less reliable orientation signals and therefore benefit more from integration with contextual information, especially if it is reliable.

Figure 2a shows the bias in target reproduction as a function of difference in flanker orientation. Both sets of stimuli show positive, assimilative effects of the flankers, with positive flanker orientation causing positive biases and negative flankers negative bias. The rounded targets show the strongest contextual effects of crowding, with peak biases varying by up to ± 5.1°, compared with ± 1.9° for the elongated targets. Furthermore, the pattern of bias follows closely that predicted and observed in serial dependence studies[33], varying non-linearly with the difference between target and flanker orientation, increasing to a maximum around ± 20°, then decreasing. These data are well fit by derivative of gaussian functions (Eq. 15, light-coloured lines), commonly used in serial dependence studies[25], and expected from a causal inference model (see modelling section[35]). The dark lines show the predictions of another Bayesian model (Eq. 10), which has also proven successful with serial dependence data[26,33]. While the models are detailed later, it is worth noting that they are almost entirely anchored by data, down to a simple scaling factor, suggesting that the data are consistent with ideal behaviour.

Another important prediction is that the contextual effects should improve performance. On a reproduction task of this sort, errors can be broadly divided into two orthogonal categories, average *accuracy* (inverse bias) and *precision* (inverse scatter about the mean value). Figure 2a reports average bias (inaccuracy), while Fig. 2b plots reproduction scatter (imprecision, given by root-variance of reproduction trials), as a function of orientation difference. As expected, at all orientation differences, scatter is lower for the elongated than the rounded targets. However, for both targets, particularly the rounded targets, the scatter decreased as the difference between target and flanker orientation decreased to be minimal when the test and flankers had matched orientation (the condition that produces maximal crowding).

To reinforce the idea that performance is at its best when flanker and target are identical, we ran a separate experiment to test two new

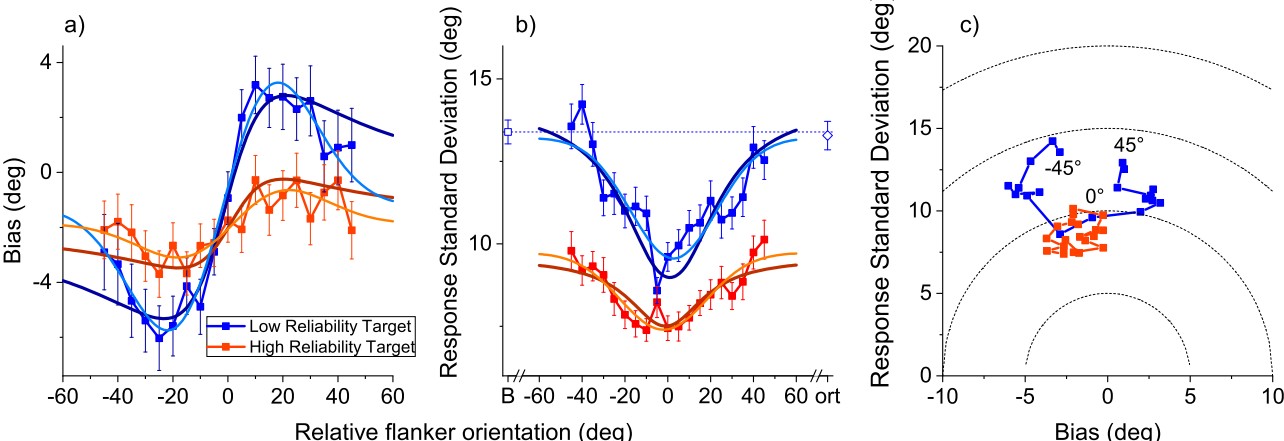

**Fig. 2 | Crowding depends on stimulus reliability and follows optimality rules. a** Average response bias (response minus target orientation) as a function of the orientation of two identical flankers. Low reliability (rounded) targets in blue, high reliability (elongated) in red. Positive biases refer to clockwise response errors, and positive orientation differences indicate that flankers are clockwise with respect to the target (so the biases are assimilative). Error bars show ±1 SEM. $N = 10$ observers. Dark lines show predictions from an ideal-observer Bayesian model which scales the action of flankers according to their reliability and orientation difference (Eq. 10 of model section). Light blue and red curves show predictions for the causal inference model that doses flanker and target information according to their reliability and the probability of originating from a common cause (Eq. 15 of model section). **b** Response standard deviation as a function of the orientation of two identical flankers, together with model predictions. Error bars show ±1 SEM. $N = 10$ observers. Colour coding as in A. Isolated squares and diamonds show results of a control experiment measuring orthogonal flankers (ort−diamonds) and an unflanked baseline (B−squares). $N = 8$ observers. **c** Total response error parsed as response standard deviation plotted against bias errors for the two conditions. Dashed circles indicate regions with identical RMS Error, given by the Pythagorean sum of the two types of error. RMSE varies with orientation, and is least around 0°, when target and flankers coincide. Source data are provided as a Source Data file.

crucial conditions in the elongated target condition: orthogonal flankers and an unflanked baseline (open diamonds and squares in Fig. 2b). Scatter for these two conditions was very similar, and more than when the flankers were present. This shows that the flankers actually improved precision, even compared with the isolated target condition.

Figure 2c plots standard deviation and bias on a two-dimensional plot, with points connected to follow the change in orientation. On this plot, total root-mean squared error is given by the Pythagorean sum of scatter and bias, the radial distance from the origin. For the points with flanker orientation most distant from the target (near ±45°), the total error is around 15°. Between these extremes, total error falls off, despite the constant bias. When the flankers and targets have similar orientations, the error falls to around 11°, evidence that "crowding" improves overall performance, by this measure.

If the effects shown in Fig. 2 represent visual crowding, they should depend on critical spacing between target and flankers, and follow Bouma's law[1]. We therefore measured the effects as a function of target-flanker spacing, for 5 participants. Figure 3 shows the data for the rounded targets with elongated flankers (which show the strongest effects). For the two smallest spacings (5.5 and 7.5 deg), bias showed the characteristic S-shaped dependency on the orientation of the flankers. For the larger spacings (11.0 and 16.6 deg), however, the effect was much reduced and even inverted at 11 deg. As before, the curves are fit by a derivative of gaussian function (Eq. 18), which is the product of a linear regression (illustrated by dashed line in Fig. 3a) and a gaussian. The best fitting slope of this regression is an estimate of the weight given to the flankers when judging orientation. Figure 3b plots the fitted weight as a function of target-flanker spacing (lower abscissa), with the upper abscissa showing the normalized target-flanker distance, the distance between target and flanker centres divided by the eccentricity (26 deg). The weight drops from 0.5 to 0 for normalized target-flanker distances between 0.3 and 0.4, broadly in line with the literature, suggesting that the effects observed here relate to crowding.

The results so far show that integration is not obligatory, but depends on the reliability of both target and flankers. They are also in line with previous studies showing that effects are maximal when

targets are most similar to flankers. A remaining question is how the flankers integrate with the target: each separately, or after combination with each other. Figure 4 illustrates two possibilities (see also modelling section). One is a feedforward model where the target integrates independently with low-level, high-resolution neural representations of each of the flankers. The other depicts integration with a broader representation including both flankers, potentially implemented through recurrent feedback.

To distinguish between these two plausible possibilities, we measured target bias with the orientation of the two flankers varying independently. Specifically, one flanker (randomly top or bottom) was always oriented +15° from the target, while the other varied randomly over the range. The logic is that the gaussian function windowing the contextual effect should be centred where the orientations of target and context coincide. If the integration occurs directly between the target and individual flankers, then the maximum effects should occur when the variable flanker coincides with the target; on the other hand, if the integration is with a broader representation including both flankers, maximum integration should occur when the flanker mean is zero, which occurs when the variable flanker is −15°. These predictions are illustrated in Fig. 4b: note that the individual flanker effect also predicts the curve to be higher at all flanker orientations, as the fixed flanker will exert a constant effect at all orientations of the variable flanker.

The results for the rounded targets with elongated flankers are shown Fig. 5a. The biases clearly follow the signature pattern, well fit by a derivative of gaussian function. The centre of the function is −10.8°, closer to the −15° predicted by integration with the average orientation of the flankers, than to 0° predicted by the individual flanker model. The mean height of the function is 0.5°, close to that observed in the previous experiment (−0.9°), while the individual-flanker integration model predicts a constant average bias 4.7°. Figure 5b shows the scatter for this experiment, which was reduced over the region of bias, well described by an inverted Gaussian with center at −14.26°, again close to the −15° predicted by the average orientation of the flankers.

To test significance, we bootstrapped the data 1000 times (sampling with replacement) and measured the centre of the gaussian

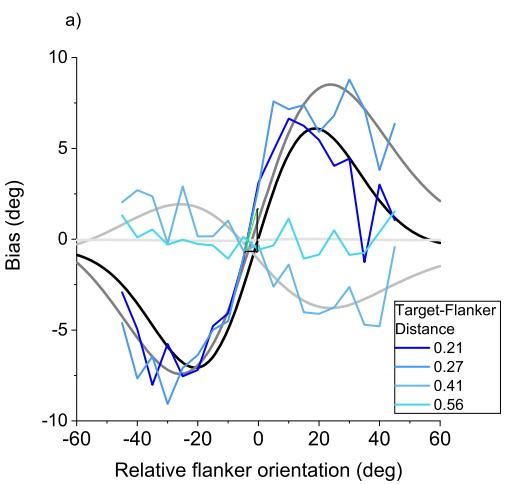

**Fig. 3 | Flanker integration decreases with distance, a signature of crowding.** **a** Response bias as function of flanker orientation for various target-flanker distances leading to four different normalized target-flanker distances (distance between flanker and target centres divided by eccentricity). Data are obtained from $N = 5$ observers and are fit with a derivative of gaussian function with free parameters (Eq. 18). **b** Weight of the flankers (maximal slope of the curves in panel **a**) as a function of the normalized target-flanker distance (colour-code as before). Negative weights imply a repulsive, rather than attractive effect of flankers on target. Error bars show ± 1 SEM. Source data are provided as a Source Data file.

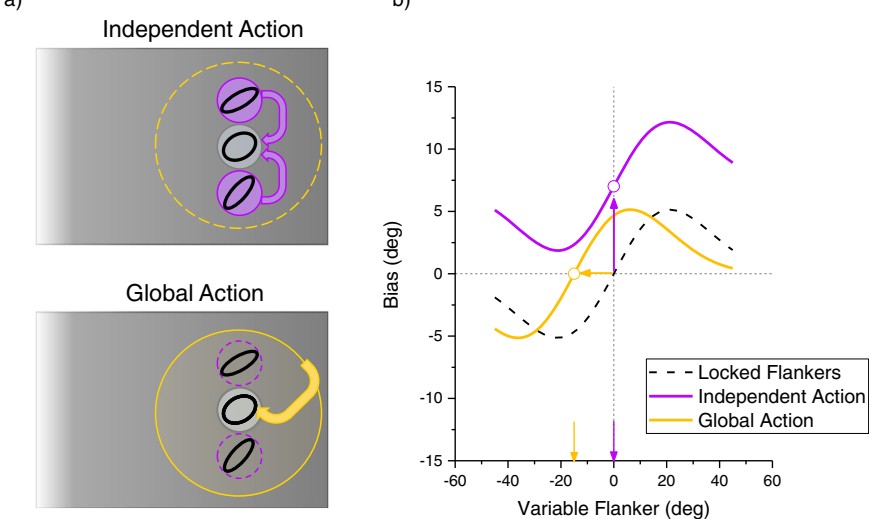

**Fig. 4 | Two possible sites of integration and their experimental predictions.** **a** Rationale for investigating the site of action of the flankers. They could either act independently on the target (illustrated by purple arrows in top left panel), or first pooled within a larger RF, which in turn biases the target (illustrated by the large yellow circle and arrow in bottom left panel). **b** Predictions for the two hypotheses. If the flankers act independently, when one is locked at + 15° and the other free to vary, the pattern should be similar to that of the main experiment (centre close to 0°), but raised because of the action of the locked flanker (purple curve). If flankers are first integrated at a more global stage, maximal effect is expected when all the elements in the larger operator average 0°. Since one of the flankers is locked at + 15°, this occurs when the other flanker is −15°, leading to a leftward shift of the curve of the main experiment (yellow curve).

derivative on each iteration. The results plotted in the histogram of Fig. 5c show that on only 16 out of 1000 iterations (1.6%) was the centre closer to 0° (individual flanker prediction) than to −15° (joint-flanker prediction). This leads to a likelihood ratio (Bayes factor) of 984/16 = 61.5, strong evidence in favour of the joint-flanker-integration model.

## Ideal observer model

We propose two plausible models to explain the pattern of data. Both are motivated by principles of "optimal cue integration" commonly used in multi-sensory perception[34,36], which predict optimal combination of information from multiple sources after appropriate weighting to minimize overall root-mean-square error. The first is based on an ideal-observer model successfully used to model serial dependence[26], the second on a "causal-inference" model of multi-sensory integration[35]. Both models predict the data well.

The ideal observer model selects the appropriate weight to assign to the flankers in order to minimize the total error in the reproduction task[26].

Total RMS error ($E$) can be decomposed into bias ($B$) and precision (scatter standard deviation: $S$), whose squares sum to give total squared error:

$$E = \sqrt{B^2 + S^2} \tag{1}$$

The ideal responses ($R$) in a pooling model can be expressed as a linear weighted combination of internal representation of target ($T$)

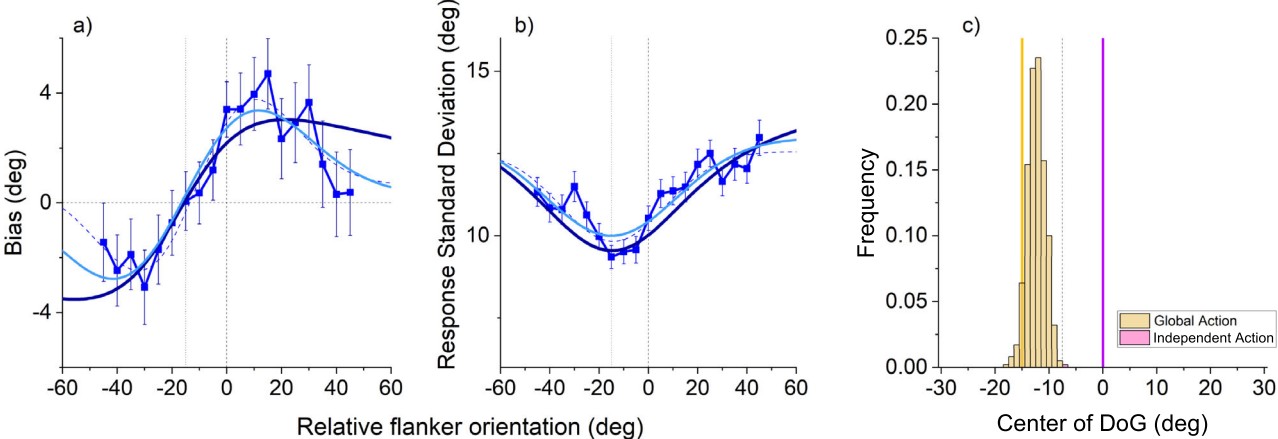

**Fig. 5 | Integration depends on orientation of both flankers, indicating a global site of action. a** Biasing errors as function of a single flanker orientation, while the other flanker was locked at + 15°. Error bars show ±1 SEM. Colours and conventions as for Fig. 2. $N = 13$ observers. Thick dark lines refer to the ideal observer model (Eq. 10), thick light blue lines to the causal inference model (Eq. 15).

Thin dashed lines show best-fitting derivative of gaussian, with all parameters free to vary (Eq. 18). **b** Response standard deviation as a function of the variable flanker orientation. Conventions as in panel **a. c** Histogram of the centres of the gaussian derivative for 1000 bootstrap fits. Source data are provided as a Source Data file.

and flankers ($F_1$ and $F_2$), each weighted by $w_1$ and $w_2$.

$$R = w_1 F_1 + w_2 F_2 + (1 - w_1 - w_2)T \qquad (2)$$

As the two flankers of this study had the same aspect ratio they should be weighted equally, ($w_1 = w_2 = w$), so Eq. 2 simplifies to:

$$R = wF_1 + wF_2 + (1 - 2w)T \qquad (3)$$

The mean of the responses ($\mu_R$) is a simple linear combination of the means of flankers and target ($\mu_1$, $\mu_2$ and $\mu_T$).

$$\mu_R = w\mu_1 + w\mu_2 + (1 - 2w)\mu_T = w(\mu_1 + \mu_2) + (1 - 2w)\mu_T \qquad (4)$$

Bias is the difference between the mean estimated response ($\mu_R$) and real orientation, $x_T$; $B = \mu_R - x_T$. Using Eq. 4 and considering that the average target representation ($\mu_T$) should be unbiased and coincide with target ($\mu_T = x_T$) it follows that:

$$B = \mu_R - x_T = w(\mu_1 + \mu_2) + \mu_T - 2w\mu_T - x_T = w(\mu_1 + \mu_2 - 2\mu_T) \qquad (5)$$

The term $\mu_1 + \mu_2 - 2\mu_T$ can be rearranged as $2((\mu_1 + \mu_2)/2 - \mu_T)$ which is twice the distance between the average of the flanker representations, $(\mu_1 + \mu_2)/2$, and the target representation $\mu_T$. For convenience we define:

$$d = (\mu_1 + \mu_2)/2 - \mu_T \qquad (6)$$

so that Eq. 5 becomes:

$$B = w(\mu_1 + \mu_2 - 2\mu_T) = 2wd \qquad (7)$$

Variance of the linear combination of the flankers and target is itself a linear combination of the flanker and target variances ($\sigma_F^2$ and $\sigma_T^2$) with the squared coefficients

$$S^2 = w^2\sigma_F^2 + w^2\sigma_F^2 + (1 - 2w)^2\sigma_T^2 \qquad (8)$$

From Eqs. 1, 7 and 8 it follows that *RMSE* can be written as:

$$E = 4w^2d^2 + w^2\sigma_F^2 + w^2\sigma_F^2 + (1 - 2w)^2\sigma_T^2 = 4w^2d^2 + w^2\sigma_F^2 \\ + w^2\sigma_F^2 + (1 - 4w + 4w^2)\sigma_T^2 \qquad (9)$$

Since RMSE is a function of second order of $w$, it is minimized when $w = \frac{-b}{2a}$, so the optimal weight ($w_{opt}$) is obtained at:

$$w_{opt} = -\frac{1}{2}\frac{-4\sigma_T^2}{4\sigma_T^2 + 2\sigma_F^2 + 4d^2} = \frac{\sigma_T^2}{2\sigma_T^2 + \sigma_F^2 + 2d^2} \qquad (10)$$

This equation has much in common with all Bayesian-like integrations used in multi-sensory research and serial dependence: the weight depends directly on *target* variance $\sigma_T^2$, so *targets* of low reliability (inverse variance) benefit more from integration, resulting in higher weighting to the *flankers*. Increase in flanker variance ($\sigma_F^2$) has the opposite effect.

The term $2d^2$ is fundamental for the signature function found in serial dependence literature, as the weight of the *flankers* will decrease with angular difference between target and average flanker orientation. This ensures that contextual cues are used only if they are plausibly similar to the target[26,32,33]. Importantly, the point that will ensure maximal weight of the flankers is when the target coincides with the average of the flankers (i.e. $d^2 = 0$).

Equations 3 and 10 define the optimal dependence of flankers. However, the flankers are not the only contextual information that may influence responses. Two obvious (and related) examples are serial dependence and "regression to the mean", both leading to a tendency to underestimate 55° and overestimate 35°. Given that the average reproduction of the two angles was 50.5° and 37.2° respectively, we assume that the regression to the mean ($\rho$) was a factor of 0.67 (calculated as (50.5−37.2)/20). Thus the final estimate of response $R$ needs to be multiplied by this factor, and a constant weighted bias summed, pulling the responses towards the average (45°). We also include an additional scaling factor ($\alpha$) comprising any further unspecified influences or sub-optimal behaviour:

$$R = \alpha\rho(wF_1 + wF_2 + (1 - 2w)T) + (1 - \rho)45 \qquad (11)$$

## Causal inference model

An alternative model prescribes that an optimal blend of information is obtained by maximum likelihood combination of two sources (assuming that the two curves originate from the same cause), multiplied by the probability that the two sources originate from the same cause[35]. Within this framework maximal interaction between cues occurs when the two sources coincide, where the weight assigned to the two cues is the well known formula used in sensory integration literature[34,36] (see also Eq. 10):

$$w_A^{max} = \frac{\sigma_B^2}{\sigma_A^2 + \sigma_B^2} \quad (12)$$

The probability of the two sources originating from a common cause can be calculated using Bayes' Theorem, as demonstrated in[35]. Assuming gaussian probability distribution functions (with centres at $\mu_A$ and $\mu_B$ and variances $\sigma_A^2$ and $\sigma_B^2$), the solution is solvable analytically[35]:

$$p(A,B|C=1) \propto \exp\left(-\frac{1}{2}\frac{(\mu_A-\mu_B)^2\sigma_P^2 + (\mu_A-\mu_P)^2\sigma_B^2 + (\mu_B-\mu_P)^2\sigma_A^2}{\sigma_A^2\sigma_B^2 + \sigma_A^2\sigma_P^2 + \sigma_B^2\sigma_P^2}\right) \quad (13)$$

where $\mu_P$ and $\sigma_P^2$) are the mean and variance of the a-prior likelihood of there being one cause (the *prior*, also gaussian). If no prior knowledge is available ($\sigma_P^2 \to \infty$) Eq. 13 simplifies to

$$p(A,B|C=1) \propto \exp\left(-\frac{1}{2}\frac{(\mu_A-\mu_B)^2}{\sigma_A^2 + \sigma_B^2}\right) \quad (14)$$

This is a gaussian peaking when the distributions of the two cues coincide ($\mu_A = \mu_B$) and falling off with a space constant related to the sum of their variances ($\sigma_A^2 + \sigma_B^2$).

In the specific case of our experiment we can map the two sources of information to the flanker compound (a gaussian with centre at $\mu_C = (\mu_1 + \mu_2)/2$, variance $\sigma_C^2 = \sigma_F^2/2$) and the target (assumed gaussian with centre $\mu_T$, and variance $\sigma_T^2$. Putting together Eqs. 12 and 14, the bias (difference between the response and the target) is given by:

$$B = w_F^{max}\, p(F,T|C=1)(\mu_F - \mu_T) = \frac{\sigma_T^2}{\sigma_C^2 + \sigma_T^2}\exp\left(-\frac{1}{2}\frac{(\mu_C-\mu_T)^2}{\sigma_C^2 + \sigma_T^2}\right)(\mu_C - \mu_T) \quad (15)$$

Which is a derivative of gaussian as a function of average flanker orientation $\mu_C$.

It also follows that response scatter is minimized only when the system considers a common cause likely (Eq. 14), predicting U-shaped (gaussian) plots of Figs. 2b and 5b.

As above, we need to incorporate regression to the mean ($\rho = 0.67$) and to allow for suboptimal behaviour to this end we introduce two free parameters that regulate the amplitude of the dependency on the flankers ($\beta$) and the breadth of the region of interaction ($\gamma$) so that the average bias is:

$$B = \beta\rho\frac{\sigma_T^2}{\sigma_C^2 + \sigma_T^2}\exp\left(-\frac{1}{2}\frac{(\mu_F-\mu_T)^2}{\gamma^2(\sigma_C^2 + \sigma_T^2)}\right)(\mu_F - \mu_T) \quad (16)$$

Interestingly, comparable behaviour is obtained if, instead of constructing a system which multiplies probabilities as in[35], one considers a system that measures the similarity between two distributions via their point-by-point product of the distributions and takes either the peak or area under the distribution.

The product of gaussians is itself a gaussian, is centred at $(\frac{\mu_B\sigma_A^2 + \mu_A\sigma_B^2}{\sigma_A^2 + \sigma_B^2})$, has variance $(\frac{\sigma_A^2\sigma_B^2}{\sigma_A^2 + \sigma_B^2})$ and peak at:

$$\frac{1}{2\pi\sigma_A\sigma_B}\exp\left(-\frac{(\mu_A-\mu_B)^2}{2(\sigma_A^2 + \sigma_B^2)}\right) \quad (17)$$

So the peak embeds the same behaviour of Eq. 14. It is easy to demonstrate that also the area under the curve follows the same gaussian dependency on the distance between cues as the area of a gaussian is equivalent to the peak (Eq. 16) times the standard deviation of the curve ($\sqrt{\frac{\sigma_A^2\sigma_B^2}{\sigma_A^2 + \sigma_B^2}}$) and a constant factor $1/\sqrt{2\pi}$ all of which are constant once the distributions have known width and thus reduce to a scaling factor.

## Model fitting

The predictions of the two modelling approaches are overlaid on the data of Figs. 2 and 5 with dark and light colours. To minimize degrees of freedom we derived the values of sensory reliability from the data of Fig. 2b, assuming that the extreme points ($\pm 30°$ and $\pm 45°$) give baseline data, not influenced by flanker integration: this is 13.2 for rounded targets (blue symbols), and 10.0 for elongated targets (red symbols).

We implemented the ideal observer model (Eq. 11) with only a scaling constant ($\alpha$), which allows for sub-optimal behaviour. These fits are particularly good for the rounded targets (with largest effects), with $R^2$ of 0.93 and 0.71 (for bias and scatter), and 0.37 and 0.74 for elongated targets) and come about assuming $\alpha = 0.7$ and 0.5 for the two conditions. One of the key features of the ideal observer model is that it reduces RMSE by leveraging on all available information. Thus it predicts the Global Integration of Fig. 4, with centres of the Gaussian derivatives close to $-15°$. Besides capturing this key feature, the model also provides good quantitative fits to the data of Fig. 5a with $R^2$ of 0.73 and average fits to those of Fig. 5b 0.76 for bias and scatter respectively ($\alpha=0.57$).

We used the same reliability values from Fig. 2b to implement the "optimal causality gating model"[35], the derivative of gaussian function plotted with light colours in Figs. 2 and 5. The sensory reliabilities fix both the maximal slope of the curve (see Eq. 12) and the width of the region of interaction (see Eq. 14). Assuming the same sensory precisions as above (13.2 and 10.0 for the two types of stimuli) maximal slopes should be 0.78 and 0.53 for the two conditions. Considering regression to the mean, which caps the possibility of detecting the weight of the flankers to about 0.7, the model predictions are 0.52 and 0.35, still larger than the real data (0.37 and 0.10). Also the widths (28.4 and 28.6) are larger than those predicted by Eq. 14 (18.8 and 19.2). For this reason we allowed two scaling factors, one enabling lower weighting of the context ($\beta$) and the other modulating the width ($\gamma$). Setting $\beta = 0.71$ and $\gamma = 1.51$ led to good fits with $R^2 = 0.97$ and 0.75 for the low reliability target (bias and scatter curves), and 0.73 and 0.84 for the high reliability target ($\beta = 0.28$ and $\gamma = 1.48$). As with the other model, the prediction in Experiment 2 is for large pooling of all available cues, thus the prediction is that of a centre at $-15°$. This model also provides good fits for response bias ($R^2 = 0.89$) and acceptable fits for response scatter ($R^2 = 0.78$, $\beta = 0.73$ and $\gamma = 0.97$).

## Discussion

The results of this study suggest an alternative interpretation of visual crowding: that it is a by-product of efficient Bayesian processes, which lead in general to improved perceptual performance, minimizing RMS error. We tested and provided evidence supporting several key predictions of this idea. Firstly, crowding, measured as flanker-induced orientation bias, was greatest when targets had the weakest orientation signals (least reliability) and flankers had the strongest, most

reliable, signals as predicted from most models of optimal cue combination[34,36]. The magnitude of the bias varied with the difference of target and flanker orientation, following the predicted non-linear pattern, increasing to a maximum of around 15°, then falling off for larger orientation differences. Importantly, the interaction of the flankers and target was associated with a reduction in response scatter (increased precision), which led to a reduction in total RMS error, an index of improved performance. Finally, the results suggest that the bias does not result from direct interactions with individual flankers, but from interaction with a representation of the average orientations of the two flankers. All these results were predicted by optimal feature combination principles, and quantitatively well modelled an ideal-observer model that minimizes reproduction errors.

Our results are consistent with previous observations showing that crowding depends on relative strength or saliency of target and flankers, with high contrast targets more immune to crowding, and configurations embedded in noise leading to stronger assimilative effects[37,38]. Our stimuli were matched for luminance and contrast, but differed in strengths of orientation signals (determined by aspect ratio), leading to different reliabilities, suggesting that the crucial variables are not low level properties but the reliabilities that they afford. Our results also agree with the myriad of experiments showing that similarities in shape (in this case orientation) cause maximum crowding[6,21,39,40], and offer an explanation of why.

These results are clearly difficult to reconcile with simple models of obligatory integration[12,41], indicating that these mechanisms are more sophisticated and selective than previously envisaged. Passive integration systems may be tweaked to explain the stronger effects for more elongated flankers (such as having more Fourier energy at that orientation), but cannot easily explain the fall off in crowding effects when the difference exceeds 15°. Any basic integrator would necessarily combine orientation energy of all angles, not only similar angles. On the other hand, the flexible integrator models proposed here (Eqs. 10 and 15) predict both the pattern and the magnitude of the results. Furthermore, the final experiment suggests that this intelligent orientation-dependent integration is unlikely to occur directly within a higher order cell itself, as the orientation-dependent integration function aligns with the average of two disparate flankers, rather than with each individual flanker. This suggests that the integration is between the target and a broad representation that includes both flankers. Mechanisms operating directly between target and individual flankers, such as the proposed "local association field"[42] between neurones of similar tuning[43], are not consistent with the results of Fig. 5, which shows that flankers are first combined with each other before exerting their effects on the target.

Combination of target and a broad representation of both flankers could be implemented in several ways. One physiologically plausible mechanism would be feedback from mid-level areas, such as V2[18,19] or V4[2,16,17], which have large receptive fields, integrating over a wide area. These cells could contain information of both flankers (as well as the target), which could be fed back to low levels (e.g. V1) to integrate flexibly with finer representations of the target. Within this framework the fine-grain target information is not lost, but combined with broad contextual information in an optimal manner to improve performance. This is analogous to the process of serial dependence, where higher-level representations of perceptual history (often termed *Bayesian priors*) are generated at mid- to high-levels of analysis, but feed back onto fairly low processing levels[44]. Similar processes could evoke crowding, integrating over space rather than time.

Interesting, the spatial selectivity of serial dependence seems to be spatiotopic, in external rather than retinal coordinates[25,45]. Crowding has also been shown to be spatiotopically selective[46,47]. Spatiotopic selectivity is a signature of high-level and functionally complex processing, indicating that both crowding and serial dependence involve moderately high levels of analysis.

Similarity between target and flankers is a major diagnostic criterion of crowding, and the current study suggests a reason for this. The interaction between target and flankers is determined by two main factors: relative *reliability* (highly related to salience) of target and flankers and, importantly, by target-flanker similarity ($d$ in Eq. 10, $(\mu_C - \mu_T)$ in Eq. 15). This explains why the biasing effects are maximal for similarly oriented flankers, and steadily fall off. Formally, this behaviour derives from theoretical minimization of total RMS errors, explained in detail in the modelling section, but readily understood intuitively. RMS Error comprises two orthogonal factors, *accuracy* (average bias) and *precision* (scatter around the mean), whose Pythagorean sum yields total error. Thus, while the contextual effects do lead to inaccuracies (biases), these are more than offset by the increased precision decrease in response scatter (Fig. 2c). Clearly, if the effects were to increase continuously with orientation, then the bias would become large, and offset the reduction in scatter, leading to increased error: integration is therefore efficient only over a limited range. Note that the efficiency-driven ideal model gives good fits simultaneous to both bias and scatter data with only one free parameter, a scaling factor. This comes out at around 0.7 after taking into account other known phenomena of orientation judgements, such as regression to the mean[48,49].

Our data and modelling also shed light on why it is usually the overall stimulus configuration promotes crowding, rather than individual flankers working in isolation. Regardless of the assumptions leading to the two models (Ideal Observer or Causal Inference), they both benefit from accumulation of relevant information, and hence the context as to be ascertained by pooling both flankers. A direct consequence of this is that it is their combined value that affects the stimulus.

Maximum crowding occurs when flankers and targets are most similar; yet our results also showed that precision is maximal when orientations coincide, seemingly contrary to a vast body of much literature reporting poor performance at that point[6,7,21,50,51]. However, this apparent discrepancy depends critically on how performance is measured. As mentioned above, our technique measured separately the *accuracy* and *precision*. Response scatter (a measure of imprecision) was lowest when the orientations coincided, as predicted by our models. RMSE (comprising both precision and accuracy) was also lowest at this point. Other standard performance measures will not necessarily show this pattern. For example, measures of "percent correct" will be poor when there is a bias, and will not be improved by high precision (minimal scatter around the incorrect bias). One crowding study that measured separately bias and precision in an orientation task (as we did) found similar results to ours, with precision highest when target and flankers coincide (see Fig. 4a in[52]). Calculations from their data show that RMSE was also lowest when orientations matched, about half that of when they differed by 45°. Their data were collected at 3.7 deg eccentricity (compared to our 26 deg), showing that the results reported here generalize to lower eccentricities. Other studies measuring standard deviation of responses do not confirm so closely our results[53], but there are differences in the paradigms used (such as using forced choice rather than reproduction techniques and other details of the display sequence).

The current experiment shows that under conditions of crowding, information about the target is not necessarily lost. This is consistent with a good deal of previous evidence (see reference[54] for review), including studies showing that it can affect the ensemble judgment[12], can cause adaptation[10] and that crowding-induced biases may not affect grasping[55]. Even more dramatic are the demonstrations that increasing flanker length[56] or adding additional flankers[22] can decrease or eliminate crowding. Our study employed simple, well controlled stimuli to allow quantitative prediction and measurement of crowding effects, similar to the studies with serial dependence studies. Thus they do not readily relate to the clever uncrowding studies of Herzog and colleagues. However, it is not difficult to envisage extensions to the

model incorporating grouping principles within the rules of integration, in the spirit of the general principles of our model: flexible, "intelligent" combination of signals, rather than a rigid integration via "rectify and sum" or similar rules[18].

We are not the first to propose that crowding may be beneficial to vision. Parkes et al.[12] suggested that compulsory integration could be a by-product of ensemble perception, the ability to judge some average, global property, such as orientation: what is lost in the individual perception may be gained by a perception of the gist of the ensemble. However, when examined closely, this idea did not hold up, as ensemble perception and crowding follow different psychophysical rules, suggesting they are different processes[57]. It would be interesting to see whether applying a similar approach to ensemble perception (asking observers to report average rather than target orientation) would help further to explore the commonalities and differences of the two processes.

Readers may find it paradoxical that we are claiming that crowding, which impacts heavily on so many fundamental aspects of daily life, such as face recognition and reading, could be considered in any sense "optimal" or "efficient". Clearly, optimality depends on what is being optimized. Our models lead to minimization of total root-mean-square error, a criterion used in many branches of engineering and science, and becoming increasingly popular in neuroscience, particular motor and perceptual research[34,58–61]. However, minimizing scatter and total error can also lead to misperceptions, or illusions such as the "ventriloquist effect"[36] or the "hollow mask illusion"[29]. Similarly, minimizing total error may be "ideal" for some basic tasks, but can lead to biasing errors that impact strongly on face and letter recognition.

Crowding occurs for many object properties, including contrast, motion and colour[8,21,53]. Here we have demonstrated clear signatures of optimality in crowding of orientation, a fundamental feature for shape perception, object recognition and reading. However, it is not certain that the effect will generalize to all forms of crowding. It would be interesting to extend the paradigm to other examples, such as colour or motion[53], to test whether crowding of these features cause the same form of optimal integration. Similarly, it would be interesting to measure integration under conditions where crowding is minimal, such as in central viewing, to test whether crowding and integration are causally related.

In summary, the current study suggests that crowding may be analogous to serial dependence, an index of predictive coding-like processes, pointing to similar function and mechanisms. As serial dependence has been shown to exploit temporal redundancies to maximize performance, crowding may also reflect similar exploitation of redundancies over space. It is worth noting that while the rules governing crowding are flexible, leading to improved performance, crowding remains completely obligatory: no effort of will or deployment of attention can allow us to resolve the crowded objects, or to ignore the contextual effects of the orientated flankers. Indeed, while our proposed pooling process is flexible and "intelligent", it remains automatic, not subject to voluntary control. This is similar to many of the experience-driven perceptual illusions, such as the "hollow mask illusion"[29]: no effort of will can cause us to see the inside of a hollow mask as concave, we always see the convex face. However, while visual crowding remains an obligatory limitation to object recognition, we conclude that like the effects of temporal context and experience, it is best understood not as a defect or bottleneck of the system, but the consequence of efficient exploitation of spatial redundancies of the natural world.

## Methods

### Participants

Experimental procedures are in line with the declaration of Helsinki and approved by the local ethics committee (*Commissione per l'Etica*

*della Ricerca*, University of Florence, n. 111 7 July 2020). Written informed consent was obtained from each participant, which included consent to process, preserve and publish the data in anonymous form.

Nineteen participants with normal or corrected-to-normal vision were recruited (aged 18–55 years, mean age = 34, 10 females).

### Stimuli

The stimuli, illustrated in Fig. 1a, were generated with Psychtoolbox for MATLAB (R2016b; MathWorks). They comprised an oval-shaped visual target flanked by oval-shaped upper and lower visual flankers, displayed 26 deg eccentric from the fixation point, with the target close to the horizontal meridian (vertical position was slightly varied from trial to trial to avoid pre-allocation of attention to the target) and flankers 5.5 deg away from the target. Both target and flankers were sketches of oval shapes, defined by 12 dark grey dots (diameter 0.3 deg, 1.4 deg inter-dots distant, 16.8 deg perimeter), presented against a uniform grey background. The target was orientated either +35° or +55° (clockwise) from the vertical, and flanker orientation randomly chosen in steps of 5° from −45° to +45° with respect to the target orientation. The two flankers were 5.5 deg from target, leading to a Bouma ratio of 0.2. We manipulated the reliability of orientation information of target and flanker stimuli by using two different aspect ratios, 2.8 (axes 3.48 and 1.23 deg) and 1.4 (axes 3.19 and 2.28 deg), illustrated in Fig. 1a. The more elongated target was always associated with more rounded flankers, and vice versa. In each experimental session of the three experiments, the two target-flanker combinations were shown both kinds of stimuli in random order.

### Procedure

Stimuli were displayed on a linearized 22″ LCD monitor (resolution 1920 × 1080 pixels, refresh rate 60 Hz). Observers were positioned 57 cm from the monitor, in a quiet room with dim lighting, and maintained fixation on a small (0.35 deg) black central dot. After a random delay from the observer initiating the trial, the stimulus was displayed for 167 ms. Then a thin rotatable white bar (0.05 × 5 deg with a gaussian profile) was presented at the fixation point with random orientation, and observers matched its orientation to that of the target by mouse control. In the first two experiments, the orientation of the two flankers was yoked, while in the third, one flanker was always +15° (clockwise) while the other varied from −45° to +45°. In the second experiment, the target-flanker distance varied, being 5.5, 7.5, 11.0 and 16.6 deg, leading to a normalized target-flanker distance of 0.21, 0.27, 0.4, 0.6.

Ten observers (6 females, mean age = 36) participated in the first experiment, five in the second (3 females, mean age = 40), thirteen in the third (7 females, mean age = 34). They contributed for a total of 10699 trials for the first experiment, 14377 for the second (spread across the four flanker-target distances) and 16574 for the last.

As a control experiment, aimed at measuring response standard deviation, we repeated the experiment probing two baseline conditions either with 90° flankers or unflanked. As in this experiment often there would have been a recognizable unflanked target we increased the number of possible target orientations so that they spanned from (22.5° to 67.5°) either clockwise from vertical or counter clockwise from vertical. Eight observers (5 females, mean age = 35) participated to this extra batch of data, contributing for a total of 889 trials.

### Data analysis

Responses occurred out from the range between 0.5 and 3 seconds after the stimulus offset were removed (for a total of 15.9% trials across the 3 experiments), as were responses with reproduction error greater than 35° (6.9% of trials).

For each target and relative orientation of the flanker, we calculated the average constant error (bias, positive meaning clockwise) and scatter (computing residuals separately for each observer and

averaging them). We then averaged the values for the two targets. Bias functions were fitted by a derivative of gaussian function, which can be considered to be a gaussian of width $s$ multiplied by a straight line of slope a [or w], which can be considered the weighting given to the flankers: 1 means the flankers are weighted equally to the target. Bias is given by:

$$B = a \cdot (\theta - m)\exp\left(-\frac{(\theta - m)^2}{s^2}\right) + b \qquad (18)$$

Where $\theta$ is orientation difference, $m$ the centre, and $b$ the vertical offset of the function. $a$, $b$ and $m$ were free to vary.

Scatter ($S$) was defined as the average root variance in each condition. The variation with orientation a gaussian function in the form:

$$S = a \cdot \exp\left(-\frac{(\theta - m)^2}{s^2}\right) + b \qquad (19)$$

Where $b$ is the baseline at high orientation differences and $a$ is the amplitude of the Gaussian. As Bias and Scatter likely originate from the same process, we yoked the parameter $s$ to best fit both curves.

### Reporting summary
Further information on research design is available in the Nature Research Reporting Summary linked to this article.

## Data availability
The processed data needed to evaluate the conclusions in the paper are available as a Source Data file. The raw data used in this study are available in the Zenodo database under accession code (10.5281/zenodo.6460723)[62]. Source data are provided with this paper.

## Code availability
The MATLAB source codes that were used to generate the datasets and analyse the results are available at a dedicated Zenodo repository (https://doi.org/10.5281/zenodo.6460723)[62].

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

## Acknowledgements

This work was supported by Horizon 2020 European Research Council Advanced Grant GenPercept No. 832813 (to D.C.B.), Italian Ministry of Education PRIN2017 Grants 2017SBCPZY, and FLAG-ERA Joint Transnational Call 2019 Grant DOMINO.

## Author contributions

G.M.C. designed the experiment and carried out data collection and analysis. G.D.E. carried out data collection and analysis. D.C.B. designed the experiment and wrote the manuscript. All authors viewed and approved the submitted version of the draft.

## Competing interests

The authors declare no competing interests.
