## [Peer Review File · Nature Communications]

Crowding results from optimal integration of visual targets with contextual informationREVIEWER COMMENTS

Reviewer #1 (Remarks to the Author):

This is an outstanding manuscript. The authors propose a novel and fascinating connection between crowding and serial dependence, two extensively studied areas of perception and cognition. They thoroughly test their idea psychophysically and with modeling. The results support the hypothesis and this will stimulate a lot of future research. I know this hypothesis will be provocative in the field, and not everyone will agree (it's a fairly contentious field), but this is a strength; the manuscript is exceptionally well balanced and approaches the issues in a most constructive way. The crowding field has been somewhat stagnant for years, and the authors' novel connection is much needed inspiration for researchers to pursue new directions. I expect this paper will motivate a great flurry of new experiments. I have a few minor suggestions below, but these are just requested clarifications, nothing major. Given the broad connections this manuscript makes across fields and the novelty of the idea and results, the manuscript certainly merits publication in Nature Comms.

Minor points:

Flanker Similarity and the (unmentioned) Diagnostic criteria for crowding. There's a nod to these diagnostic criteria (eg, Whitney & Levi, 2011) but not a direct statement. The critical spacing one is key of course (p. 6)—and the model does a great job predicting that—but it's not the only one. Another key characteristic directly addressed in the MS is similarity. This is relevant here bc the prior literature had little explanation for why "similarity" matters in the way that it does (eg similarity modulates crowding and dissimilarity releases crowding). The authors' idea of a connection between serial dependence and crowding, and their model, is very powerful and important in part bc it provides that "why". In future work it will be interesting to test if other diagnostic criteria like inner-outer flanker asymmetry, upper lower visual field diff, etc also hold. This isn't necessary here but readers may wonder and the authors could prompt that question and help motivate the important follow up research.

P2, "tasks like or face recognition" delete "or" and perhaps add a reference here. Maybe Farzin et al 2009 (for faces) or the cited reviews (if this is a generic statement about objects).

P3, "qualitatively and qualitatively". Perhaps one of these was intended to be "quantitatively"?

Fig 1b. Using red outline and blue outline around the respective panels (or at least red and blue color somehow in those two panels) would help readers follow the correspondence between all the Figs; red always indicates high reliability target. Might as well start using that rule in fig 1b.

P.4. "...Formally the modeling section" should be "formally in the..."

P5. "...With difference between..." should have "the" or an "s" after "difference"

Fig 2, abscissa. Add clarification that this axis is "difference" in orientation. It's not absolute orientation, right?

Fig 2. Where would isolated (single) targets be on this graph?

P.11. Is the "signature" of the "signature function" the derivative-of-Gaussian shape in the SD literature? If so, perhaps mention that or explain what is meant by "signature"

P 12. The first sentence of the "causal inference model" section. That first sentence is too difficult to parse or understand. Not just because the word "form" probably isn't intended. Rephrasing could help a lot.

P12. "...the weight assigned of is the..."Rephrase, please.

Aside from these very minor points, this is an excellent manuscript.

Reviewer #2 (Remarks to the Author):

This is a fascinating manuscript, with novel findings that present a new perspective on a widely studied phenomenon. The authors examine visual crowding, the disruptive effect of clutter on object recognition. A large body of research has depicted this effect as the 'fundamental bottleneck on object recognition' in peripheral vision especially. As a result, we know a great deal about the way this process affects object recognition and the potential mechanisms. What is much less clear is why crowding occurs in the first place. This manuscript presents an interesting answer to this question by considering its usefulness.

The broad approach here is to compare crowding with properties of 'serial dependence', the effect whereby judgements of a stimulus are influenced by the presentation of other stimuli in prior trials. With this comparison, the authors ask whether crowding can be considered to be an efficient/optimal process, rather than reflecting a disruptive bottleneck. Several predictions are made in this case, all of which are ultimately argued to be supported by the data. The authors ask observers to judge the orientation of shapes made from an outline of dots. They first observe greater biases and higher response scatter with "low reliability" near-circular target stimuli that are more difficult to judge, compared with "high reliability" elliptical targets. Second, they note that response scatter is greatest when the orientation of the flankers is close to the target, with a decrease as dissimilarity decreases (that is, performance improves as crowding increases, showing its efficiency). Finally, the pattern of biases follows the mean orientation of flankers rather than an independent combination, which is used to justify a higher-level model. The manuscript is well written and engaging, and presents a provocative view of a widely studied process. If the findings here are true then this presents an important aspect of our understanding. I do have a number of issues with the manuscript as it stands, however.

1. The pattern of response scatter

The main issue concerns the second finding – that response scatter is greatest (i.e. performance is worst) when the orientation of the flankers is most similar to the target. This finding is a key aspect of the proposal that crowding is efficient/optimal, since errors decrease as the strength of crowding increases. If true however, this finding is inconsistent with a large literature on the effect of target-flanker similarity in crowding. More typically, crowding is greatest when target and flanker elements are most similar to one another, decreasing as their dissimilarity increases (the opposite of the current observation). This has been found for a range of stimulus properties including contrast polarity, color, spatial frequency, and direction (Kooi, Toet, Tripathy, & Levi, 1994; Chung, Levi, & Legge, 2001; Gheri, Morgan, & Solomon, 2007), and in particular for orientation judgements (Andriessen & Bouma, 1976; Wilkinson, Wilson, & Ellemberg, 1997), similar to those used in the present study. In those latter studies, flankers that share similar orientations to the target induce the most crowding, with less crowding as the orientation of the flankers rotates away. Given that a major premise of the current study rests on the opposite finding, this discrepancy needs explanation and/or further exploration. The authors do in fact cite some of these studies to begin the manuscript, describing the patterns above, but the discrepancy with the current results is subsequently ignored. How can this discrepancy be explained, and how does this fit with the central arguments regarding the efficiency/optimality of crowding?

There seem to me at least two possibilities to explain the discrepancy. One is that the authors have not fully measured the range of possible target-flanker differences in orientation. Targets are

presented at either 35 or 55 degrees rotation, with flankers that differ from these values by up to ± 45 degrees. Response scatter peaks at the highest values measured (± 45 degrees). It is however possible then that these values may drop again as the differences further increase, up to their maximum of 90 degrees from the target orientation. It is typically these 90 degree values that are compared in order to show target-flanker similarity effects (Andriessen & Bouma, 1976; Wilkinson, Wilson, & Elleberg, 1997), and I suspect that if the measurements continued here that performance would drop again. Indeed – patterns of this nature have been reported in a prior study (Solomon, Felisberti, & Morgan, 2004). There, orientation sensitivity is high when flankers are similarly oriented to the target, drops for orientations up to ± 45 degrees, and then increases again as the rotation continues to 90 degrees. The same may be true of the present stimuli if a larger range of orientations were tested. Would that not alter the interpretations regarding the efficiency of this process?

A second possibility is the eccentricity – the authors present their stimuli 26 degrees from fixation. Prior observations of target-flanker similarity have tended to use lower eccentricities. Given that some properties of crowding change with eccentricity, e.g. the response biases (Mareschal, Morgan, & Solomon, 2010) and of course the well-known effects of spatial extent (Bouma, 1970), it could be that the present results are something that only arises in the far periphery. Were this the case, however, the question remains – why is efficiency evident in the present results and not these other studies?

To summarize, the results regarding response scatter appear to follow the opposite pattern to a range of well-established and replicated findings in the literature. The premise of the paper rests heavily on this observation. The authors need to demonstrate that this pattern is reliable by extending the range of their measurements in some way and/or by addressing this discrepancy with prior results. If the current results show efficiency, what does that say about all of these other results? If crowding is only efficient in these limited circumstances, is it really efficient?

2. The lack of an unflanked baseline

Part of the issue of interpretation with the above response scatter data also relates to the lack of an unflanked baseline. Typically, performance with flankers is used to measure crowding, with an unflanked baseline (with an isolated element) used to measure uncrowded performance. The authors here take their baseline using flanked performance, using flankers with orientations at the extremes of their range (p14). Given the odd pattern of response scatter (as above), this assumption is problematic. If unflanked performance is more like the values with a flanker difference of 0 degrees, then performance would go from largely unaffected with the 0 degree flankers to impaired with the ± 45 degree flankers, rather than from improved with the 0 degree flankers to unaffected with the ± 45 degree flankers, as the authors argue. Does that not change the interpretation of the results and their efficiency/optimality substantially?

3. The distinction with 'low-level' pooling models of crowding

The authors contrast their findings with 'low level pooling models', which do not seem to me to be at odds with the present results. This distinction begins in the abstract (e.g. on line 2), where low level models are contrasted with their 'alternative hypothesis', and continues throughout, e.g. on p15 of the discussion, where it is argued that pooling models cannot explaining the effects of flanker orientation. Later on however, the authors describe their model as a pooling process (p17). The mechanism proposed for their model, with large receptive fields in areas like V4 (p16), also sounds very similar to ideas raised in various pooling models. For instance, processes of 'population pooling' have attributed the crowding of orientation signals to pooling within receptive fields in area V2 or V4 (van den Berg, Roerdink, & Cornelissen, 2010; Harrison & Bex, 2015). Similar arguments are also made by 'high dimensional' pooling models (Rosenholtz, Yu, & Keshvari, 2019). In fact, patterns of bias that are very similar to those in Figure 2A of this manuscript have been reported previously and accounted for via pooling processes (Greenwood & Parsons, 2020). In this latter case, the model accounts for target-flanker similarity effects via variations in the weights applied to the flankers. I don't see why this is

inconsistent with the variations shown in the current study.

The authors distinguish between two possible models, both of which seem like pooling models to me. One is a 'low level' version in which the interactions happen independently between each flanker and the target, linked with a feedforward local process. The other involves a broader integration in which the flankers have a combined influence on the target, linked with recurrent feedback interactions. The latter does not seem wholly distinct from the operation of the most recent population pooling models (Harrison & Bex, 2015; Greenwood & Parsons, 2020) described above however. In those cases, the flankers affect the target through their combination within a single population response. My feeling is that the results of Experiment 2 in the present study would be entirely consistent with these models – when flanker orientations vary independently, their combined population response would have a shifted mean that would tend to alter the subsequent judgements related to the target. If so, then I do not think these results are inconsistent with pooling, nor do they provide clear evidence for feedback. This is not to take away from the novelty of these findings, however – I agree that the results provide clear evidence that the flankers do not interact independently with the target. The distinction between the two models presented here is certainly interesting, but their physiological basis is clearly overstated.

4. Effects of target reliability

The first result reported in the manuscript is that biases are greater and response scatter higher with "low reliability" near-circular target stimuli that are more difficult to judge, compared with "high reliability" elliptical targets. This effect is attributed to reliability, and explained via a Bayesian framework. Its relation to similar results with alternative explanations is unexplored, however. Most notably, crowding is strongest with flankers of high luminance contrast (Chung, Levi, & Legge, 2001; Pelli, Palomares, & Majaj, 2004). Lowering the target contrast can also increase crowding (Felisberti, Solomon, & Morgan, 2005). Assimilative biases related to orientation judgements are also increased when noise is added to stimuli (Mareschal, Morgan, & Solomon, 2010). Can all of these effects be understood via reliability? It seems to me there is an alternative explanation that crowding is determined by the strength of the target signal, relative to the strength of the flanker signal(s). Could these effects, including those of the present study, be understood as signal strength rather than reliability per se?

5. The relation to serial dependence

Much is made of the similarities between serial dependence and crowding, which I agree is a fascinating link to make. The arguments for efficiency in this context also sound to me like arguments made more broadly in vision for the principles of redundancy reduction (Attneave, 1954), including for processes like adaptation (Clifford, 2002) and surround suppression (Rao & Ballard, 1999). Could the similarities here in fact indicate a broader link in the form of a "canonical computation" across all of visual perception? I wonder if the strong link to serial dependence is a little short-sighted in this sense.

5. The neural basis of crowding

The idea of crowding relating to higher cortical areas like V4 is attributed to Pelli & Tillman (p2), but this idea derives from earlier work (Motter & Simoni, 2007; Motter, 2009). Others have also linked crowding with receptive field sizes in areas like V2 (He, Wang, & Fang, 2019).

6. Stimulus details

Was the rotation of flankers taken from the target orientation on each trial, such that the ± 45 degree range differed in terms of absolute orientations for the 35 and 55 degree targets?

Additionally, can we be sure that the judgements made by observers concern the orientation of these stimuli, rather than another property? Given the dotted nature of the stimuli used in the present task, perhaps observers are not judging orientation, but rather another property like the position of the outermost dots in the elements. This could allow a kind of relative position or Vernier judgement. Prior studies have tended to use line elements or Gabors in this context – if true, could this explain the difference with the studies of target-flanker similarity described above?

References

- Andriessen, J. J., & Bouma, H. (1976). Eccentric vision: Adverse interactions between line segments. *Vision Research*, 16(1), 71-78.
- Attneave, F. (1954). Some informational aspects of visual perception. *Psychological Review*, 61(3), 183-193.
- Bouma, H. (1970). Interaction effects in parafoveal letter recognition. *Nature*, 226, 177-178.
- Chung, S. T. L., Levi, D. M., & Legge, G. E. (2001). Spatial-frequency and contrast properties of crowding. *Vision Research*, 41, 1833-1850.
- Clifford, C. W. G. (2002). Perceptual adaptation: Motion parallels orientation. *Trends in Cognitive Sciences*, 6(3), 136-143.
- Felisberti, F. M., Solomon, J. A., & Morgan, M. J. (2005). The role of target salience in crowding. *Perception*, 34(7), 823-833.
- Gheri, C., Morgan, M. J., & Solomon, J. A. (2007). The relationship between search efficiency and crowding. *Perception*, 36(12), 1779-1787.
- Greenwood, J. A., & Parsons, M. J. (2020). Dissociable effects of visual crowding on the perception of color and motion. *Proceedings of the National Academy of Sciences of the United States of America*, 117(14), 8196-8202.
- Harrison, W. J., & Bex, P. J. (2015). A Unifying Model of Orientation Crowding in Peripheral Vision. *Current Biology*, 25(24), 3213-3219.
- He, D., Wang, Y., & Fang, F. (2019). The critical role of V2 population receptive fields in visual orientation crowding. *Current Biology*, 29(13), 2229-2236. e2223.
- Kooi, F. L., Toet, A., Tripathy, S. P., & Levi, D. M. (1994). The effect of similarity and duration on spatial interaction in peripheral vision. *Spatial Vision*, 8(2), 255-279.
- Mareschal, I., Morgan, M. J., & Solomon, J. A. (2010). Cortical distance determines whether flankers cause crowding or the tilt illusion. *Journal of Vision*, 10(8):13, 1-14.
- Motter, B. C. (2009). Central V4 Receptive Fields Are Scaled by the V1 Cortical Magnification and Correspond to a Constant-Sized Sampling of the V1 Surface. *Journal of Neuroscience*, 29(18), 5749-5757.
- Motter, B. C., & Simoni, D. A. (2007). The roles of cortical image separation and size in active visual search performance. *Journal of Vision*, 7(2(6)), 1-15.
- Pelli, D. G., Palomares, M., & Majaj, N. J. (2004). Crowding is unlike ordinary masking: Distinguishing feature integration from detection. *Journal of Vision*, 4(12), 1136-1169.
- Rao, R. P. N., & Ballard, D. H. (1999). Predictive coding in the visual cortex: A functional interpretation of some extra-classical receptive-field effects. *Nature Neuroscience*, 2(1), 79-87.
- Rosenholtz, R., Yu, D., & Keshvari, S. (2019). Challenges to pooling models of crowding: Implications for visual mechanisms. *Journal of Vision*, 19(7), 1-25.
- Solomon, J. A., Felisberti, F. M., & Morgan, M. J. (2004). Crowding and the tilt illusion: Toward a unified account. *Journal of Vision*, 4, 500-508.
- van den Berg, R., Roerdink, J. B. T. M., & Cornelissen, F. W. (2010). A Neurophysiologically Plausible Population Code Model for Feature Integration Explains Visual Crowding. *PLoS Computational Biology*, 6(1), e1000646.
- Wilkinson, F., Wilson, H. R., & Ellemberg, D. (1997). Lateral interactions in peripherally viewed texture arrays. *Journal of the Optical Society of America. A, Optics, Image Science, and Vision*, 14(9), 2057-2068.

Reviewer #3 (Remarks to the Author):

Cicchini and colleagues put forward the hypothesis that crowding is results from Bayes-optimal integration of visual targets with spatial context. The authors identify four features of their empirical data that are consistent with Bayes-optimal integration: (1) Crowding is strongest for reliable flankers and unreliable targets, (2) Crowding depends on flanker-target similarity (here orientation), (3) precision of orientation judgments increases with increasing flanker-target similarity, and (4) Crowding depends on similarity of targets to average flanker orientation, not individual flanker orientations. The authors present two ideal observer models (a Bayesian ideal observer, and a causal inference model), which can reproduce the above features of the empirical data.

While I find the hypothesis that crowding results from optimal integration intriguing, I am somewhat reserved when it comes to the evidence provided in the current study. I am also not convinced that the behavioral benefits described here can be ascribed to crowding rather than ensemble perception. Please find my detailed points below.

1. The ideal observer models only provide adequate fits when equipped with scaling parameters that account for "sub-optimal" behavior. The required scaling is not negligible, rescaling the optimal integration weights by ~ 40 to 50%. Therefore, it is not clear whether the observers' behavior is at all optimal, beyond resembling some qualitative features of the data. The authors could make a much stronger case when quantitatively accounting for the sub-optimal behavior. For instance, how strong would regression to the mean of orientation judgments need to be (put forward by the authors as an explanation of sub-optimality) in order to match this scaling? Is this consistent with the empirical data?

2. Related to point 1, it is not clear in how far the empirical features are exclusively accounted for by Bayes-optimal integration versus other forms of (non-optimal) integration. As the authors note in their discussion feature 1 (orientation uncertainty) could be captured by obligatory integration models. Feature 2 (flanker-target similarity) could be explained by interference between similarly tuned, and therefore more strongly interconnected neural populations. For feature 4 (global vs local context), it seems that one could develop an alternative optimal observer that integrates local instead of global context. That is, I do not understand which optimality consideration would strictly dictate global versus local integration.

I believe most of these concerns of whether the behavioral features really arise from optimal integration could be mitigated by improving point 1 above, i.e. providing a more detailed quantitative explanation of behavior, rather than absorbing a considerable mismatch between predictions and data into one or two unexplained "sub-optimality" parameters.

3. It is not clear whether the behavioral benefits examined in this experiment are due to crowding or ensemble perception. While these appear to be at least partially distinct phenomena, they can co-occur ("Reexamining the possible benefits of visual crowding: dissociating crowding from ensemble percepts" Bulakowski et al., 2011). Cicchini et al. test the influence of target-distractor distance, and demonstrate that bias depends on distance, as expected for a crowding effect. However, I would contest that increasing flanker-target distance also alters the ensemble, and can therefore also impact ensemble perception. Perhaps one way to address this issue would be to test whether or not similar integration effects occur for more foveally presented stimuli, i.e. in the absence of crowding (albeit under matched conditions of visual uncertainty). If they do, the current observations would perhaps be better explained as resulting from ensemble perception, while crowding merely co-occurs in the current setup.

Minor comments:

Figure 5B. Minimum scatter appears to occur at 0 deg, while the ideal observer models predict the

minimum to occur at 15 degrees. I am curious whether the authors have any explanation/speculation of why the bias and variance data diverge in this aspect.

In their introduction the authors state "Crowding impacts on many important daily tasks, such as face recognition and reading [...]" I would be curious how the authors reconcile this view that crowding appears to negatively impact perception in real world scenarios ("daily tasks") with their optimal integration theory.

GENERAL

We thank the editor and particularly the reviewers for their time and very helpful advice. We have taken all the suggestions on board, definitely resulting in an improved manuscript. We trust it is now acceptable for publication.

We have marked the changes in blue on the revised manuscript.

David Burr

For the authors

REVIEWER COMMENTS

Reviewer #1 (Remarks to the Author):

This is an outstanding manuscript. The authors propose a novel and fascinating connection between crowding and serial dependence, two extensively studied areas of perception and cognition. They thoroughly test their idea psychophysically and with modeling. The results support the hypothesis and this will stimulate a lot of future research. I know this hypothesis will be provocative in the field, and not everyone will agree (it's a fairly contentious field), but this is a strength; the manuscript is exceptionally well balanced and approaches the issues in a most constructive way. The crowding field has been somewhat stagnant for years, and the authors' novel connection is much needed inspiration for researchers to pursue new directions. I expect this paper will motivate a great flurry of new experiments. I have a few minor suggestions below, but these are just requested clarifications, nothing major. Given the broad connections this manuscript makes across fields and the novelty of the idea and results, the manuscript certainly merits publication in Nature Comms.

We thank the reviewer for their kind words, and agree that the findings will be contentious, but hopefully stimulate useful research.

Minor points:

Flanker Similarity and the (unmentioned) Diagnostic criteria for crowding. There's a nod to these diagnostic criteria (eg, Whitney & Levi, 2011) but not a direct statement. The critical spacing one is key of course (p. 6)—and the model does a great job predicting that—but it's not the only one. Another key characteristic directly addressed in the MS is similarity. This is relevant here bc the prior literature had little explanation for why "similarity" matters in the way that it does (eg similarity modulates crowding and dissimilarity releases crowding). The authors' idea of a connection between serial dependence and crowding, and their model, is very powerful and important in part bc it provides that "why". In future work it will be interesting to test if other diagnostic criteria like inner-outer flanker asymmetry, upper lower visual field diff, etc also hold. This isn't necessary here but readers may wonder and the authors could prompt that question and help motivate the important follow up research.

Thanks for this important suggestion. We now mention the diagnosis criteria more clearly in introduction, and pick it up again in discussion.

P2, "tasks like or face recognition" delete "or" and perhaps add a reference here. Maybe Farzin et al 2009 (for faces) or the cited reviews (if this is a generic statement about objects).

P3, "qualitatively and qualitatively". Perhaps one of these was intended to be "quantitatively"?

Fig 1b. Using red outline and blue outline around the respective panels (or at least red and blue color somehow in those two panels) would help readers follow the correspondence between all the Figs; red always indicates high reliability target. Might as well start using that rule in fig 1b.

Excellent idea

P.4. "...Formally the modeling section" should be "formally in the..."

P5. "...With difference between..." should have "the" or an "s" after "difference"

Fig 2, abscissa. Add clarification that this axis is "difference" in orientation. It's not absolute orientation, right?

Fig 2. Where would isolated (single) targets be on this graph?

Unfortunately we did not measure the effects without flankers.

P.11. Is the “signature” of the “signature function” the derivative-of-Gaussian shape in the SD literature? If so, perhaps mention that or explain what is meant by “signature”

P 12. The first sentence of the “causal inference model” section. That first sentence is too difficult to parse or understand. Not just because the word “form” probably isn’t intended. Rephrasing could help a lot.

P12. “...the weight assigned of is the...”Rephrase, please.

Aside from these very minor points, this is an excellent manuscript.

Thank you very much, all the minor points have all been dealt with.

Reviewer #2 (Remarks to the Author):

This is a fascinating manuscript, with novel findings that present a new perspective on a widely studied phenomenon. The authors examine visual crowding, the disruptive effect of clutter on object recognition. A large body of research has depicted this effect as the ‘fundamental bottleneck on object recognition’ in peripheral vision especially. As a result, we know a great deal about the way this process affects object recognition and the potential mechanisms. What is much less clear is why crowding occurs in the first place. This manuscript presents an interesting answer to this question by considering its usefulness.

Thank you for the kind words. Thank you also for the very detailed help you have given, providing useful references and encouraging us to make clearer our ideas.

The broad approach here is to compare crowding with properties of ‘serial dependence’, the effect whereby judgements of a stimulus are influenced by the presentation of other stimuli in prior trials. With this comparison, the authors ask whether crowding can be considered to be an efficient/optimal process, rather than reflecting a disruptive bottleneck. Several predictions are made in this case, all of which are ultimately argued to be supported by the data. The authors ask observers to judge the orientation of shapes made from an outline of dots. They first observe greater biases and higher response scatter with “low reliability” near-circular target stimuli that are more difficult to judge, compared with “high reliability” elliptical targets. Second, they note that response scatter is greatest when the orientation of the flankers is close to the target, with a decrease as dissimilarity decreases (that is, performance improves as crowding increases, showing its efficiency). Finally, the pattern of biases follows the mean orientation of flankers rather than an independent combination, which is used to justify a higher-level model. The manuscript is well written and engaging, and presents a provocative view of a widely studied process. If the findings here are true then this presents an important aspect of our understanding. I do have a number of issues with the manuscript as it stands, however.

1. The pattern of response scatter

The main issue concerns the second finding – that response scatter is greatest (i.e. performance is worst) when the orientation of the flankers is most similar to the target. This finding is a key aspect of the proposal that crowding is efficient/optimal, since errors decrease as the strength of crowding increases. If true however, this finding is inconsistent with a large literature on the effect of target-flanker similarity in crowding. More typically, crowding is greatest when target and flanker elements are most similar to one another, decreasing as their dissimilarity increases (the opposite of the current observation). This has been found for a range of stimulus properties including contrast polarity, color, spatial frequency, and direction (Kooi, Toet, Tripathy, & Levi, 1994; Chung, Levi, & Legge, 2001; Gheri, Morgan, & Solomon, 2007), and in particular for orientation judgements (Andriessen & Bouma, 1976; Wilkinson, Wilson, & Ellemberg, 1997), similar to those used in the present study. In those latter studies, flankers that share similar orientations to the target induce the most crowding, with less crowding as the orientation of the flankers rotates away. Given that a major premise of the current study rests on the opposite finding, this discrepancy needs explanation and/or further exploration. The authors do in fact cite some of these studies to begin the manuscript, describing the patterns above, but the discrepancy with the current results is subsequently ignored. How can this discrepancy be explained, and how does this fit with the central arguments regarding the efficiency/optimality of crowding?

This important comment shows that we need to do a better job explaining our results and ideas. Firstly, we assume that the first sentence was a typo – response scatter is *least* (precision greatest) when orientations coincide (where crowding is *greatest*). That may seem counter-intuitive, but it depends on how you measure crowding. Typically it is percent correct, or perhaps contrast sensitivity. We are saying that RMS Errors (which comprise both accuracy and precision) are reduced, because although average accuracy decreases (strong bias), the increased precision more than offsets the bias, resulting in lower RMSE (radial distance in Fig 2C). However, other performance measures need not necessarily improve. For example, simple measures of accuracy (whether the reproduction was near veridical) would be low when the bias is high, as the orientation

judgement would seldom be “correct”; improved precision would not increase accuracy, and possibly decrease it, all responses become more tightly grouped around the incorrect bias. The only paper to measure separately bias and precision that we know of is Solomon, Felisberti and Morgan (JoV 2004: thanks for the pointer), and they report results very similar to ours (their Figure 4A). Also their data indicate a reduction in RMS Error (although they did not describe their results that way), shown in this figure below derived from their data. We have tried to make the explanations clearer in the results and discussion sections, and have added a brief paragraph discussing the apparent paradox.

Figure 1: RMSE calculated from Figure 4A of Solomon et al., 2004

There seem to me at least two possibilities to explain the discrepancy. One is that the authors have not fully measured the range of possible target-flanker differences in orientation. Targets are presented at either 35 or 55 degrees rotation, with flankers that differ from these values by up to ± 45 degrees. Response scatter peaks at the highest values measured (± 45 degrees). It is however possible then that these values may drop again as the differences further increase, up to their maximum of 90 degrees from the target orientation. It is typically these 90 degree values that are compared in order to show target-flanker similarity effects (Andriessen & Bouma, 1976; Wilkinson, Wilson, & Ellemberg, 1997), and I suspect that if the measurements continued here that performance would drop again. Indeed – patterns of this nature have been reported in a prior study (Solomon, Felisberti, & Morgan, 2004). There, orientation sensitivity is high when flankers are similarly oriented to the target, drops for orientations up to ± 45 degrees, and then increases again as the rotation continues to 90 degrees. The same may be true of the present stimuli if a larger range of orientations were tested. Would that not alter the interpretations regarding the efficiency of this process?

See above response for the explanation of apparent discrepancy. It is interesting that precision (but not bias) improved for 90° flankers in Solomon et al. Unfortunately we did not measure out that far. However, it would not change our story, as we are interested in the range where the flankers cause crowding by biasing results; in that range there is a clear trade-off between accuracy and precision for both our and their data. Our data show maximum bias at a lower orientation than theirs (about 20° compared with their 45°), so it did not seem necessary to measure beyond 45°.

A second possibility is the eccentricity – the authors present their stimuli 26 degrees from fixation. Prior observations of target-flanker similarity have tended to use lower eccentricities. Given that some properties of crowding change with eccentricity, e.g. the response biases (Mareschal, Morgan, & Solomon, 2010) and of course the well-known effects of spatial extent (Bouma, 1970), it could be that the present results are something that only arises in the far periphery. Were this the case, however, the question remains – why is efficiency evident in the present results and not these other studies?

It is true that we used a large eccentricity, to maximize the effects, but our results agree with the only other study where efficiency can be calculated (Solomon et al, above figure), who worked at the much lower eccentricity of 3.7° (now mentioned).

To summarize, the results regarding response scatter appear to follow the opposite pattern to a range of well-established and replicated findings in the literature. The premise of the paper rests heavily on this observation. The authors need to demonstrate that this pattern is reliable by extending the range of their measurements in some way and/or by addressing this discrepancy with prior results. If the current results show efficiency, what does that say about all of these other results? If crowding is only efficient in these limited circumstances, is it really efficient?

We hope the above comments address this seeming paradox to the reviewer's satisfaction.

2. The lack of an unflanked baseline

Part of the issue of interpretation with the above response scatter data also relates to the lack of an unflanked baseline. Typically, performance with flankers is used to measure crowding, with an unflanked baseline (with an isolated element) used to measure uncrowded performance. The authors here take their baseline using flanked performance, using flankers with orientations at the extremes of their range (p14). Given the odd pattern of response scatter (as above), this assumption is problematic. If unflanked performance is more like the values with a flanker difference of 0 degrees, then performance would go from largely unaffected with the 0 degree flankers to impaired with the ± 45 degree flankers, rather than from improved with the 0 degree flankers to unaffected with the ± 45 degree flankers, as the authors argue. Does that not change the interpretation of the results and their efficiency/optimality substantially?

Unfortunately we did not measure a baseline. We did not think this is a major problem, but in retrospect it would have been useful to do so.

3. The distinction with 'low-level' pooling models of crowding

The authors contrast their findings with 'low level pooling models', which do not seem to me to be at odds with the present results. This distinction begins in the abstract (e.g. on line 2), where low level models are contrasted with their 'alternative hypothesis', and continues throughout, e.g. on p15 of the discussion, where it is argued that pooling models cannot explaining the effects of flanker orientation. Later on however, the authors describe their model as a pooling process (p17). The mechanism proposed for their model, with large receptive fields in areas like V4 (p16), also sounds very similar to ideas raised in various pooling models. For instance, processes of 'population pooling' have attributed the crowding of orientation signals to pooling within receptive fields in area V2 or V4 (van den Berg, Roerdink, & Cornelissen, 2010; Harrison & Bex, 2015). Similar arguments are also made by 'high dimensional' pooling models (Rosenholtz, Yu, & Keshvari, 2019). In fact, patterns of bias that are very similar to those in Figure 2A of this manuscript have been reported previously and accounted for via pooling processes (Greenwood & Parsons, 2020). In this latter case, the model accounts for target-flanker similarity effects via variations in the weights applied to the flankers. I don't see why this is inconsistent with the variations shown in the current study.

The authors distinguish between two possible models, both of which seem like pooling models to me. One is a 'low level' version in which the interactions happen independently between each flanker and the target, linked with a feedforward local process. The other involves a broader integration in which the flankers have a combined influence on the target, linked with recurrent feedback interactions. The latter does not seem wholly distinct from the operation of the most recent population pooling models (Harrison & Bex, 2015; Greenwood & Parsons, 2020) described above however. In those cases, the flankers affect the target through their combination within a single population response. My feeling is that the results of Experiment 2 in the present study would be entirely consistent with these models – when flanker orientations vary independently, their combined population response would have a shifted mean that would tend to alter the subsequent judgements related to the target. If so, then I do not think these results are inconsistent with pooling, nor do they provide clear evidence for feedback. This is not to take away from the novelty of these findings, however – I agree that the results provide clear evidence that the flankers do not interact independently with the target. The distinction between the two models presented here is certainly interesting, but their physiological basis is clearly overstated.

Indeed most models of crowding since Morgan's Nature paper involve some sort of obligatory pooling. The two novelty of what we propose are that the pooling is "intelligent", occurring when it leads to improved efficiency (measured by RMS Error); and that it is relatable to serial dependence, allowing cross-fertilization of the two fields. No previous model that we are aware of predicts this. Certainly there are elements in common, especially with our preferred model involving compulsory (unintelligent) pooling of signals; but the key difference is that this integrated information is not to final output but is combined intelligently with the more local signal. We have tried to make that clearer in the text.

4. Effects of target reliability

The first result reported in the manuscript is that biases are greater and response scatter higher with "low reliability" near-circular target stimuli that are more difficult to judge, compared with "high reliability" elliptical targets. This effect is attributed to reliability, and explained via a Bayesian framework. Its relation to similar results with alternative explanations is unexplored, however. Most notably, crowding is strongest with flankers of high luminance contrast (Chung, Levi, & Legge, 2001; Pelli, Palomares, & Majaj, 2004). Lowering the target contrast can also increase crowding (Felisberti, Solomon, & Morgan, 2005). Assimilative biases related to orientation judgements are also increased when noise is added to stimuli (Mareschal, Morgan, & Solomon, 2010). Can all of these effects be understood via reliability? It seems to me there is an alternative explanation that crowding is determined by the strength of the target signal, relative to the strength of the

flanker signal(s). Could these effects, including those of the present study, be understood as signal strength rather than reliability per se?

This is a good point, and we have now added those references. Technically, *reliability* is the inverse of the variance of the underlying noise distribution. Variance will certainly be affected by contrast and by noise in the way suggested (these have been the standard techniques of manipulating reliability in the multi-sensory literature), so yes, signal strength will increase reliability. However, given that our modelling is quantitatively based on reliability of flanker and target, we prefer to remain within that framework.

It is also interesting to note that the stimuli we employ (ellipses defined by dots), on the other hand allow manipulating reliability while keeping more basic parameters (contrast, visibility etc) as matched as possible. This suggests that the framework we chose could be a more general one and could encompass those special cases reported in the literature

5. The relation to serial dependence

Much is made of the similarities between serial dependence and crowding, which I agree is a fascinating link to make. The arguments for efficiency in this context also sound to me like arguments made more broadly in vision for the principles of redundancy reduction (Attneave, 1954), including for processes like adaptation (Clifford, 2002) and surround suppression (Rao & Ballard, 1999). Could the similarities here in fact indicate a broader link in the form of a “canonical computation” across all of visual perception? I wonder if the strong link to serial dependence is a little short-sighted in this sense.

This is a great idea, but we feel it goes beyond this paper. We would prefer to link crowding to a specific and well studied phenomenon, like serial dependence, rather than trying to push our claims too far at this stage. But we expect that the idea of canonical calculation will prove to be correct.

5. The neural basis of crowding

The idea of crowding relating to higher cortical areas like V4 is attributed to Pelli & Tillman (p2), but this idea derives from earlier work (Motter & Simoni, 2007; Motter, 2009). Others have also linked crowding with receptive field sizes in areas like V2 (He, Wang, & Fang, 2019).

Thanks for this feedback, we now have acknowledged these earlier scholars.

6. Stimulus details

Was the rotation of flankers taken from the target orientation on each trial, such that the ± 45 degree range differed in terms of absolute orientations for the 35 and 55 degree targets?

Yes it was.

Additionally, can we be sure that the judgements made by observers concern the orientation of these stimuli, rather than another property? Given the dotted nature of the stimuli used in the present task, perhaps observers are not judging orientation, but rather another property like the position of the outermost dots in the elements. This could allow a kind of relative position or Vernier judgement. Prior studies have tended to use line elements or Gabors in this context – if true, could this explain the difference with the studies of target-flanker similarity described above?

If we understand correctly the reviewer is asking us to consider the possibility that observers are not judging orientation of the main axis of the ellipse but rather the relative position of the outermost dot of the target object. This quantity could provide a rough proxy for orientation in that when the target exceeds (i.e. it is more to the right) the flankers likely the target is more horizontal (and vice versa if more inmost). If this was the case however in the “rounded target – slim flankers” the target would not exceed the flankers in this metric and reports should lean towards vertical. Conversely in the “slim target – rounded flanker”, condition. These two predictions however are not met as there are no substantial biases.

Another possibility is that observers were judging the relative orientation (or position) of the two outmost dots of each stimulus. This would enable for instance judging the absolute orientation of the object (whereas previous hypothesis only would inform on relative orientation respect to the flanker). However, this mechanism cannot account for the clear difference of precision between the two types of stimuli.

References

- Andriessen, J. J., & Bouma, H. (1976). Eccentric vision: Adverse interactions between line segments. *Vision Research*, 16(1), 71-78.
- Attneave, F. (1954). Some informational aspects of visual perception. *Psychological Review*, 61(3), 183-193.
- Bouma, H. (1970). Interaction effects in parafoveal letter recognition. *Nature*, 226, 177-178.
- Chung, S. T. L., Levi, D. M., & Legge, G. E. (2001). Spatial-frequency and contrast properties of crowding. *Vision Research*, 41, 1833-1850.
- Clifford, C. W. G. (2002). Perceptual adaptation: Motion parallels orientation. *Trends in Cognitive Sciences*, 6(3), 136-143.
- Felisberti, F. M., Solomon, J. A., & Morgan, M. J. (2005). The role of target salience in crowding. *Perception*, 34(7), 823-833.
- Gheri, C., Morgan, M. J., & Solomon, J. A. (2007). The relationship between search efficiency and crowding. *Perception*, 36(12), 1779-1787.
- Greenwood, J. A., & Parsons, M. J. (2020). Dissociable effects of visual crowding on the perception of color and motion. *Proceedings of the National Academy of Sciences of the United States of America*, 117(14), 8196-8202.
- Harrison, W. J., & Bex, P. J. (2015). A Unifying Model of Orientation Crowding in Peripheral Vision. *Current Biology*, 25(24), 3213-3219.
- He, D., Wang, Y., & Fang, F. (2019). The critical role of V2 population receptive fields in visual orientation crowding. *Current Biology*, 29(13), 2229-2236. e2223.
- Kooi, F. L., Toet, A., Tripathy, S. P., & Levi, D. M. (1994). The effect of similarity and duration on spatial interaction in peripheral vision. *Spatial Vision*, 8(2), 255-279.
- Mareschal, I., Morgan, M. J., & Solomon, J. A. (2010). Cortical distance determines whether flankers cause crowding or the tilt illusion. *Journal of Vision*, 10(8):13, 1-14.
- Motter, B. C. (2009). Central V4 Receptive Fields Are Scaled by the V1 Cortical Magnification and Correspond to a Constant-Sized Sampling of the V1 Surface. *Journal of Neuroscience*, 29(18), 5749-5757.
- Motter, B. C., & Simoni, D. A. (2007). The roles of cortical image separation and size in active visual search performance. *Journal of Vision*, 7(2(6)), 1-15.
- Pelli, D. G., Palomares, M., & Majaj, N. J. (2004). Crowding is unlike ordinary masking: Distinguishing feature integration from detection. *Journal of Vision*, 4(12), 1136-1169.
- Rao, R. P. N., & Ballard, D. H. (1999). Predictive coding in the visual cortex: A functional interpretation of some extra-classical receptive-field effects. *Nature Neuroscience*, 2(1), 79-87.
- Rosenholtz, R., Yu, D., & Keshvari, S. (2019). Challenges to pooling models of crowding: Implications for visual mechanisms. *Journal of Vision*, 19(7), 1-25.
- Solomon, J. A., Felisberti, F. M., & Morgan, M. J. (2004). Crowding and the tilt illusion: Toward a unified account. *Journal of Vision*, 4, 500-508.
- van den Berg, R., Roerdink, J. B. T. M., & Cornelissen, F. W. (2010). A Neurophysiologically Plausible Population Code Model for Feature Integration Explains Visual Crowding. *PLoS Computational Biology*, 6(1), e1000646.
- Wilkinson, F., Wilson, H. R., & Ellemberg, D. (1997). Lateral interactions in peripherally viewed texture arrays. *Journal of the Optical Society of America. A, Optics, Image Science, and Vision*, 14(9), 2057-2068.

Reviewer #3 (Remarks to the Author):

Cicchini and colleagues put forward the hypothesis that crowding is results from Bayes-optimal integration of visual targets with spatial context. The authors identify four features of their empirical data that are consistent with Bayes-optimal integration: (1) Crowding is strongest for reliable flankers and unreliable targets, (2) Crowding depends on flanker-target similarity (here orientation), (3) precision of orientation judgments increases with increasing flanker-target similarity, and (4) Crowding depends on similarity of targets to average flanker orientation, not individual flanker orientations. The authors present two ideal observer models (a Bayesian ideal observer, and a causal inference model), which can reproduce the above features of the empirical data.

While I find the hypothesis that crowding results from optimal integration intriguing, I am somewhat reserved when it comes to the evidence provided in the current study. I am also not convinced that the behavioral benefits described here can be ascribed to crowding rather than ensemble perception. Please find my detailed points below.

1. The ideal observer models only provide adequate fits when equipped with scaling parameters that account for "sub-optimal" behavior. The required scaling is not negligible, rescaling the optimal integration weights by ~40 to 50%. Therefore, it is not clear whether the observers' behavior is at all optimal, beyond resembling some qualitative features of the data. The authors could make a much stronger case when quantitatively accounting for the sub-optimal behavior. For instance, how strong would regression to the mean of orientation judgments need to be (put forward by the authors as an explanation of sub-optimality) in order to match this scaling? Is this consistent with the empirical data?

Thanks for this important suggestion. We were trying to keep our models as simple as possible to be more accessible to the reader, but the calculation proposed is quite simple. Regression to the mean compresses the output by about 30%, which accounts for much of the underestimation. We now need a scaling factor of only 0.7 to fit the data. The main features of the models is that the same models that fit well serial dependence capture the DoG pattern of results (to a scaling factor), but we agree it is more impressive when the scaling factor is close to unity.

2. Related to point 1, it is not clear in how far the empirical features are exclusively accounted for by Bayes-optimal integration versus other forms of (non-optimal) integration. As the authors note in their discussion feature 1 (orientation uncertainty) could be captured by obligatory integration models. Feature 2 (flanker-target similarity) could be explained by interference between similarly tuned, and therefore more strongly interconnected neural populations. For feature 4 (global vs local context), it seems that one could develop an alternative optimal observer that integrates local instead of global context. That is, I do not understand which optimality consideration would strictly dictate global versus local integration.

I believe most of these concerns of whether the behavioral features really arise from optimal integration could be mitigated by improving point 1 above, i.e. providing a more detailed quantitative explanation of behavior, rather than absorbing a considerable mismatch between predictions and data into one or two unexplained "sub-optimality" parameters.

We trust that the improved absolute fit goes some way towards addressing the referee's concerns. We do of course accept that mechanisms of the type suggested could be involved, such as interconnections between similarly tuned neural populations (which we now mention, pointing out they are not consistent with the second experiment). However, we believe that our models put considerable constraints on how these mechanisms act. We also believe that it is constructive to relate crowding to serial dependence, which is currently being studied intensely, prompting cross-fertilization of ideas and discussions between the two important fields of research.

3. It is not clear whether the behavioral benefits examined in this experiment are due to crowding or ensemble perception. While these appear to be at least partially distinct phenomena, they can co-occur ("Reexamining the possible benefits of visual crowding: dissociating crowding from ensemble percepts" Bulakowski et al., 2011). Cicchini et al. test the influence of target-distractor distance, and demonstrate that bias depends on distance, as expected for a crowding effect. However, I would contest that increasing flanker-target distance also alters the ensemble, and can therefore also impact ensemble perception. Perhaps one way to address this issue would be to test whether or not similar integration effects occur for more foveally presented stimuli, i.e. in the absence of crowding (albeit under matched conditions of visual uncertainty). If they do, the current observations would perhaps be better explained as resulting from ensemble perception, while crowding merely co-occurs in the current setup.

Thanks for pointing us to this interesting paper, of which we were unaware. It is interesting that ensemble perception and crowding follow partially different rules, opening the way for interesting experiments. In our experiment we only measured crowding (judging orientation of a single target, rather than the average). It would certainly be interesting to do the converse (ensemble judgements) to see if the two phenomena followed similar rules. If they do not, it would be further proof that the two are at least partially different phenomena. This is of course a large new study (which the second author may pursue for his PhD), but we do now mention ensemble perception, and cite the important study mentioned. Thank you

Minor comments:

Figure 5B. Minimum scatter appears to occur at 0 deg, while the ideal observer models predict the minimum to occur at 15 degrees. I am curious whether the authors have any explanation/speculation of why the bias and variance data diverge in this aspect.

Indeed, the prediction was off. However, we had previously calculated variance of the aggregate participant together, so different individual biases added artificially to the variance. We now remove the individual biases from the calculation of scatter (essentially calculating variance separately for each participant and averaging), and the result is far closer to the predicted 15° (see new figure 5B). We explain this procedure in the methods.

In their introduction the authors state "Crowding impacts on many important daily tasks, such as face recognition and reading [...]" I would be curious how the authors reconcile this view that crowding appears to negatively impact perception in real world scenarios ("daily tasks") with their optimal integration theory.

This is a very good point, thank you. Perhaps it goes a bit beyond the scope of this study, but we now add a paragraph discussing how optimizing for one aspect (minimal RMS Errors) may lead to sub-optimal behaviour of others (such as face recognition), as occurs in other forms of Bayesian optimization.

REVIEWER COMMENTS

Reviewer #1 (Remarks to the Author):

The authors have thoroughly addressed all of the reviewer concerns, and the manuscript is acceptable for publication.

Reviewer #2 (Remarks to the Author):

As in the first submission, this is a fascinating manuscript, with novel findings that present a new perspective on the widely studied phenomenon of crowding. The revisions have done much to clarify the contributions of this work and the nature of the associated analyses. My major issue was previously with the findings regarding response scatter and their relation to prior work, which is now addressed to some extent in the revised manuscript. Many of the other issues have also been resolved. However, there remains one major issue in particular that has been completely ignored in the revisions, along with some minor issues. It is clear to me that this paper presents a novel and useful viewpoint to the literature, but it is especially important that these claims be supported by evidence. At the moment it is still not clear to me that this is the case.

1. The lack of an unflanked baseline

The major shortcoming with the revised manuscript is the lack of an unflanked performance baseline – the authors seek to measure crowding (recognition of a target when surrounded by flankers) but never measure performance in the absence of crowding (a target without flankers). This issue was raised in the first submission, to which the authors responded that it “would have been useful”, but this shortcoming was neither measured nor is its absence acknowledged in the revised manuscript. This absence leads to problematic assumptions about the data, a potentially problematic implementation within the models, and problematic statements about the nature of crowding.

The authors assert at many points that “crowding improves overall performance” (p6), and that they observe “improved perceptual performance” (p16), “a reduction in response scatter [and] total RMS error” (p16), “to improve performance” (p17) and “improved performance” (p20). However, without the measurement of an unflanked baseline where crowding is absent (i.e. measurement of orientation perception for an isolated target without flankers), it is not clear what this improvement is relative to.

In the absence of this measurement, the authors have assumed that the most extreme values of target-flanker difference that were tested correspond to the performance baseline. As noted in my first review, this is not at all clear. Prior studies have shown that the reduction in crowding with increased dissimilarity between target and flankers does not often return performance to the unflanked baseline. This can be seen for instance in the cited studies on orientation judgements (Wilkinson, Wilson, & Elleberg, 1997; Solomon, Felisberti, & Morgan, 2004) – a target surrounded by orthogonal flankers is not recognized as well as a target presented on its own. In the data of Wilkinson et al, for instance, the threshold elevations (from unflanked) can remain as high as 2.5 times the unflanked baseline, even with dissimilar flanker elements. It is not possible to assume that crowding is absent so long as the flankers remain present, in other words.

If the authors were able to demonstrate that the response scatter with similar flankers (near the middle of Figure 2b) is indeed better than unflanked performance, this would be a convincing demonstration that the presence of crowding improves performance relative to its absence (with a target in isolation). Otherwise, what we are looking at is an improvement in performance when crowding is strong vs. when it is reduced. A far more problematic outcome would be that the unflanked baseline may in fact yield very low values of response scatter, which would correspond more closely to the values with similar flankers (near the middle). In this case, all of the performance

with flankers would in fact be a decrement/impairment, rather than an improvement. In other words, the apparent benefits of crowding that are described here would simply be a case of the authors arbitrarily relabelling "up" as "down". Without these measurements, we can never know. The question of optimality is never truly addressed until we do.

This is not simply a quibble about stimulus details – it is a key aspect of the measurements being performed here. Without these measurements, the way in which performance with these circumstances is unclear, as reflected in the lack of a meaningful performance comparison in the statements quoted above (i.e. is it that crowding improves performance relative to uncrowded performance, or simply that strong crowding is better than reduced crowding?).

These assumptions regarding performance also carry through to the modelling. On p15 the authors use the extreme points of the dataset as a baseline to estimate optimal sensory reliability. This in turn carries through to key operations of the model (equation 12). If the unflanked reliability is in fact lower than this point, can crowding truly be said to be optimal? Precision (when calculated as in this study) is certainly better when crowded biases are strongest (with small target-flanker differences) compared to when they are reduced, but the authors cannot say whether this is optimal compared to the absence of crowding altogether.

2. Assumptions in the model and optimality

Much has been improved in the discussion of how these findings relate to prior reports of impaired performance in tasks like letter and face recognition (p19). It seems to me however that a key aspect of this discrepancy relates to the assumptions of the model. The text on p11 states that "ideal responses...can be expressed as a linear weighted combination of [the] target and flankers...". But this is surely suboptimal when the task is to judge the orientation of the target and to ignore the flankers (as in prior studies on letter/face recognition etc). The updated description of the causal inference model makes this assumption more explicit where it is stated (p13) that "...an optimal blend [assumes] that the two curves originate from the same cause" (with a similar comment on p14). With this assumption in place, I could see how these interactions are optimal, and indeed given the structural regularities of the visual scene (that similar orientations, colors, etc are likely to be found together) this is perhaps a sensible assumption for the most part, as exploited by texture/statistical models of peripheral vision and crowding (Freeman & Simoncelli, 2011; Rosenholtz, Yu, & Keshvari, 2019). But it is problematic in tasks where the observer must recognize the target, ignoring the flankers. Although performance may then be optimal given this assumption, it is clearly not optimal given the task being required of observers in (most) crowding paradigms. In these cases, where the task is clearly very different (identify the target face amongst flankers, for instance), is there not therefore an over-application of this principle and thus a suboptimality? I suspect that this again relates back to the lack of an unflanked baseline in the authors' measurements, and the missing perspective that arises from this.

3. The findings regarding response scatter

The pattern of results obtained with response scatter are much more clear now. However, the authors now describe the discrepancy between the current and previous studies as being due to their separate measurement of bias and precision (p18). The work of Solomon, Felisberti, and Morgan (2004) is cited in this context as 'the only study to our knowledge' to have similarly measured bias and precision. A range of other studies have used this approach however, some of which report patterns that do not quite fit with that observed here, to my eye. For instance, Glen and Dakin (2013) report sensitivity and biases for orientation crowding, while Greenwood and Parsons (2020) measured crowded biases and precision/thresholds for color and motion. The patterns there do not quite match those of the current study, with the least precision arising when flankers are most similar to the target, though it could indeed be the case that on the whole the combination of bias and precision leads to a pattern of response scatter similar to that of the current work.

4. Relation to pooling models

The discussion of the relationship to pooling models is much improved in the introduction and modelling sections. But there is still a statement in the discussion (p16) that the findings are 'difficult to reconcile' with pooling models. On the contrary, effects of target-flanker similarity are simulated by pooling models both in Greenwood and Parsons (2020) and in the context of texture pooling models (Rosenholtz, Yu, & Keshvari, 2019). Again, these approaches do not seem so dissimilar to that employed in the current study.

5. Missing references

At points the authors make reference to prior literature without citing the studies being referred to, particularly in the new additions to the manuscript. This occurs on p2 ("Crowding is stronger in the upper than the lower visual field, and for radial than for tangential flankers") and p16 ("the myriad of experiments showing that similarities in shape...cause maximum crowding"). It is particularly unclear to me what studies the latter refers to.

6. Biases

It would help the clarity of the manuscript to have the direction of biases explained somewhere, e.g. on p6 to explain that the errors in Figure 2 follow the orientation of the flankers, and that they presumably go in the opposite direction with some separations in Figure 3.

References

- Freeman, J., & Simoncelli, E. P. (2011). Metamers of the ventral stream. *Nature Neuroscience*, 14, 1195-1201.
- Glen, J. C., & Dakin, S. C. (2013). Orientation-crowding within contours. *Journal of Vision*, 13, 1-11.
- Greenwood, J. A., & Parsons, M. J. (2020). Dissociable effects of visual crowding on the perception of color and motion. *Proceedings of the National Academy of Sciences of the United States of America*, 117, 8196-8202.
- Rosenholtz, R., Yu, D., & Keshvari, S. (2019). Challenges to pooling models of crowding: Implications for visual mechanisms. *Journal of Vision*, 19, 1-25.
- Solomon, J. A., Felisberti, F. M., & Morgan, M. J. (2004). Crowding and the tilt illusion: Toward a unified account. *Journal of Vision*, 4, 500-508.
- Wilkinson, F., Wilson, H. R., & Ellemberg, D. (1997). Lateral interactions in peripherally viewed texture arrays. *Journal of the Optical Society of America. A, Optics, Image Science, and Vision*, 14, 2057-2068.

Reviewer #3 (Remarks to the Author):

I am still not entirely sure whether *Nature Communications* is the best outlet, as the study puts forward an intriguing idea, but leaves open many questions that would require more data. However, I agree with the authors and fellow reviewers that this is an interesting research direction.

I appreciate the authors' attempt to minimize the scaling factor, by quantitatively accounting for known suboptimalities due to the oblique bias. I am still not entirely convinced that we are dealing with optimal integration, or whether part of the required scaling is due to suboptimal integration, which would limit the authors' claim of crowding resulting from optimal integration. If not done in the current manuscript, this will be an important point for future studies.

I realize that I have been somewhat unclear about my previous point about crowding vs ensemble

perception. My main question was whether the spatial integration observed in the current experiments could perhaps be more general than just occurring for peripheral targets surrounded by flankers (i.e., the conditions under which crowding occurs). For instance, would a similar bias occur for very noisy *foveal* target stimuli surrounded by flankers, in the absence of crowding. That is, would observers generally rely on a weighted average of an ensemble, when visual information about the target is very poor. In the current experiments, crowding might increase the uncertainty/reliability of the target, which might prompt observers to integrate information of the spatial context. In this case, optimal integration would be the *consequence* of crowding, not the *cause* of crowding. If I understand the authors correctly, they claim that optimal integration is the cause of the crowding phenomenon. I would appreciate if the authors could clarify this point.

I also concur Reviewer #2 that an unflanked baseline (or orthogonal flankers) would help to clarify whether maximally similar flankers indeed enhance performance (better than baseline), or whether they are least detrimental (equal or worse performance than baseline).

Please find enclosed the response to the reviewers and the novel submission with all changes flagged in blue.

Reviewer #1 (Remarks to the Author):

The authors have thoroughly addressed all of the reviewer concerns, and the manuscript is acceptable for publication.

Reviewer #2 (Remarks to the Author):

As in the first submission, this is a fascinating manuscript, with novel findings that present a new perspective on the widely studied phenomenon of crowding. The revisions have done much to clarify the contributions of this work and the nature of the associated analyses. My major issue was previously with the findings regarding response scatter and their relation to prior work, which is now addressed to some extent in the revised manuscript. Many of the other issues have also been resolved. However, there remains one major issue in particular that has been completely ignored in the revisions, along with some minor issues. It is clear to me that this paper presents a novel and useful viewpoint to the literature, but it is especially important that these claims be supported by evidence. At the moment it is still not clear to me that this is the case.

1. The lack of an unflanked baseline

The major shortcoming with the revised manuscript is the lack of an unflanked performance baseline – the authors seek to measure crowding (recognition of a target when surrounded by flankers) but never measure performance in the absence of crowding (a target without flankers). This issue was raised in the first submission, to which the authors responded that it “would have been useful”, but this shortcoming was neither measured nor is its absence acknowledged in the revised manuscript. This absence leads to problematic assumptions about the data, a potentially problematic implementation within the models, and problematic statements about the nature of crowding.

We have now measured baselines for the rounded targets condition (the main one, that leads to the strongest effects). We also measure with orthogonal flankers. The results are shown in Figure 2 as hollow squares/diamonds. The two thresholds are similar to each other, and worse than all the other flanker conditions. We hope this is sufficient for publication now.

The authors assert at many points that “crowding improves overall performance” (p6), and that they observe “improved perceptual performance” (p16), “a reduction in response scatter [and] total RMS error” (p16), “to improve performance” (p17) and “improved performance” (p20). However, without the measurement of an unflanked baseline where crowding is absent (i.e. measurement of

orientation perception for an isolated target without flankers), it is not clear what this improvement is relative to.

In the absence of this measurement, the authors have assumed that the most extreme values of target-flanker difference that were tested correspond to the performance baseline. As noted in my first review, this is not at all clear. Prior studies have shown that the reduction in crowding with increased dissimilarity between target and flankers does not often return performance to the unflanked baseline. This can be seen for instance in the cited studies on orientation judgements (Wilkinson, Wilson, & Elleberg, 1997; Solomon, Felisberti, & Morgan, 2004) – a target surrounded by orthogonal flankers is not recognized as well as a target presented on its own. In the data of Wilkinson et al, for instance, the threshold elevations (from unflanked) can remain as high as 2.5 times the unflanked baseline, even with dissimilar flanker elements. It is not possible to assume that crowding is absent so long as the flankers remain present, in other words.

If the authors were able to demonstrate that the response scatter with similar flankers (near the middle of Figure 2b) is indeed better than unflanked performance, this would be a convincing demonstration that the presence of crowding improves performance relative to its absence (with a target in isolation). Otherwise, what we are looking at is an improvement in performance when crowding is strong vs. when it is reduced. A far more problematic outcome would be that the unflanked baseline may in fact yield very low values of response scatter, which would correspond more closely to the values with similar flankers (near the middle). In this case, all of the performance with flankers would in fact be a decrement/impairment, rather than an improvement. In other words, the apparent benefits of crowding that are described here would simply be a case of the authors arbitrarily relabelling “up” as “down”. Without these measurements, we can never know. The question of optimality is never truly addressed until we do.

This is not simply a quibble about stimulus details – it is a key aspect of the measurements being performed here. Without these measurements, the way in which performance with these circumstances is unclear, as reflected in the lack of a meaningful performance comparison in the statements quoted above (i.e. is it that crowding improves performance relative to uncrowded performance, or simply that strong crowding is better than reduced crowding?).

These assumptions regarding performance also carry through to the modelling. On p15 the authors use the extreme points of the dataset as a baseline to estimate optimal sensory reliability. This in turn carries through to key operations of the model (equation 12). If the unflanked reliability is in fact lower than this point, can crowding truly be said to be optimal? Precision (when calculated as in this study) is certainly better when crowded biases are strongest (with small target-flanker differences) compared to when they are reduced, but the authors cannot say whether this is optimal compared to the absence of crowding altogether.

We accept your arguments, thank you, and have added the baseline to the main condition. We agree that this strengthens the manuscript.

2. Assumptions in the model and optimality

Much has been improved in the discussion of how these findings relate to prior reports of impaired performance in tasks like letter and face recognition (p19). It seems to me however that a key aspect of this discrepancy relates to the assumptions of the model. The text on p11 states that “ideal responses...can be expressed as a linear weighted combination of [the] target and flankers...”. But this is surely suboptimal when the task is to judge the orientation of the target and to ignore the flankers (as in prior studies on letter/face recognition etc). The updated description of the causal inference model makes this assumption more explicit where it is stated (p13) that “...an optimal blend [assumes] that the two curves originate from the same cause” (with a similar comment on p14). With this assumption in place, I could see how these interactions are optimal, and indeed given the structural regularities of the visual scene (that similar orientations, colors, etc are likely to be found together) this is perhaps a sensible assumption for the most part, as exploited by texture/statistical models of peripheral vision and crowding (Freeman & Simoncelli, 2011; Rosenholtz, Yu, & Keshvari, 2019). But it is problematic in tasks where the observer must recognize the target, ignoring the flankers. Although performance may then be optimal given this assumption, it is clearly not optimal given the task being required of observers in (most) crowding paradigms. In these cases, where the task is clearly very different (identify the target face amongst flankers, for instance), is there not therefore an over-application of this principle and thus a suboptimality? I suspect that this again relates back to the lack of an unflanked baseline in the authors’ measurements, and the missing perspective that arises from this.

We now stress that optimization of basic features while optimal strictly speaking, may still impact negatively in higher recognition processes. Indeed it is not uncommon that optimal processes lead to illusions such as the ventriloquist effect and the hollow face illusion

3. The findings regarding response scatter

The pattern of results obtained with response scatter are much more clear now. However, the authors now describe the discrepancy between the current and previous studies as being due to their separate measurement of bias and precision (p18). The work of Solomon, Felisberti, and Morgan (2004) is cited in this context as ‘the only study to our knowledge’ to have similarly measured bias and precision. A range of other studies have used this approach however, some of which report patterns that do not quite fit with that observed here, to my eye. For instance, Glen and Dakin (2013) report sensitivity and biases for orientation crowding, while Greenwood and Parsons (2020) measured crowded biases and precision/thresholds for color and motion. The patterns there do not quite match those of the current study, with the least precision arising when flankers are most similar to the target, though it could indeed be the case that on the whole the combination of bias and precision leads to a pattern of response scatter similar to that of the current work.

Thank you, we were not aware of these studies. We now reference them and point out the differences in results (possibly due to differences in the experimental techniques). But we also take the opportunity to add a penultimate paragraph mentioning that our results may not generalize

beyond orientation, encouraging experiments along these lines for other features such as motion and color. Thank you for prompting this caveat.

4. Relation to pooling models

The discussion of the relationship to pooling models is much improved in the introduction and modelling sections. But there is still a statement in the discussion (p16) that the findings are 'difficult to reconcile' with pooling models. On the contrary, effects of target-flanker similarity are simulated by pooling models both in Greenwood and Parsons (2020) and in the context of texture pooling models (Rosenholtz, Yu, & Keshvari, 2019). Again, these approaches do not seem so dissimilar to that employed in the current study.

Thank you. We have toned down our claims of novelty, but do still want to stress the major difference, of flexible pooling.

5. Missing references

At points the authors make reference to prior literature without citing the studies being referred to, particularly in the new additions to the manuscript. This occurs on p2 ("Crowding is stronger in the upper than the lower visual field, and for radial than for tangential flankers") and p16 ("the myriad of experiments showing that similarities in shape...cause maximum crowding"). It is particularly unclear to me what studies the latter refers to.

Thank you we now added relevant references

6. Biases

It would help the clarity of the manuscript to have the direction of biases explained somewhere, e.g. on p6 to explain that the errors in Figure 2 follow the orientation of the flankers, and that they presumably go in the opposite direction with some separations in Figure 3.

Thank you we now do.

References

Freeman, J., & Simoncelli, E. P. (2011). Metamers of the ventral stream. *Nature Neuroscience*, 14, 1195-1201.

Glen, J. C., & Dakin, S. C. (2013). Orientation-crowding within contours. *Journal of Vision*, 13, 1-11.

Greenwood, J. A., & Parsons, M. J. (2020). Dissociable effects of visual crowding on the perception of color and motion. *Proceedings of the National Academy of Sciences of the United States of America*, 117, 8196-8202.

Rosenholtz, R., Yu, D., & Keshvari, S. (2019). Challenges to pooling models of crowding: Implications for visual mechanisms. *Journal of Vision*, 19, 1-25.

Solomon, J. A., Felisberti, F. M., & Morgan, M. J. (2004). Crowding and the tilt illusion: Toward a unified account. *Journal of Vision*, 4, 500-508.

Wilkinson, F., Wilson, H. R., & Ellemberg, D. (1997). Lateral interactions in peripherally viewed texture arrays. *Journal of the Optical Society of America. A, Optics, Image Science, and Vision*, 14, 2057-2068.

Reviewer #3 (Remarks to the Author):

I am still not entirely sure whether *Nature Communications* is the best outlet, as the study puts forward an intriguing idea, but leaves open many questions that would require more data. However, I agree with the authors and fellow reviewers that this is an interesting research direction.

I appreciate the authors' attempt to minimize the scaling factor, by quantitatively accounting for known suboptimalities due to the oblique bias. I am still not entirely convinced that we are dealing with optimal integration, or whether part of the required scaling is due to suboptimal integration, which would limit the authors' claim of crowding resulting from optimal integration. If not done in the current manuscript, this will be an important point for future studies.

I realize that I have been somewhat unclear about my previous point about crowding vs ensemble perception. My main question was whether the spatial integration observed in the current experiments could perhaps be more general than just occurring for peripheral targets surrounded by flankers (i.e., the conditions under which crowding occurs). For instance, would a similar bias occur for very noisy foveal target stimuli surrounded by flankers, in the absence of crowding. That is, would observers generally rely on a weighted average of an ensemble, when visual information about the target is very poor. In the current experiments, crowding might increase the uncertainty/reliability of the target, which might prompt observers to integrate information of the spatial context. In this case, optimal integration would be the consequence of crowding, not the cause of crowding. If I understand the authors correctly, they claim that optimal integration is the cause of the crowding phenomenon. I would appreciate if the authors could clarify this point.

We thank the referee for making their point more clear. It is certainly a good point, which we are not able to address at this stage (other than showing in Figure 3 that the assimilative effects disappear at large separations). We do now mention that this is an open question meriting further research (in the penultimate paragraph).

I also concur Reviewer #2 that an unflanked baseline (or orthogonal flankers) would help to clarify whether maximally similar flankers indeed enhance performance (better than baseline), or whether they are least detrimental (equal or worse performance than baseline).

We too were convinced, and did the extra measurements (open symbols in Figure 2).

REVIEWERS' COMMENTS

Reviewer #2 (Remarks to the Author):

The authors have responded comprehensively to issues raised in the previous rounds of submission, and my major issues with the manuscript are resolved. The new baseline measurements (with an additional condition including orthogonally-oriented flankers) are a convincing addition, providing a clear reference for the arguments that crowding can improve the precision of the shape judgements measured by the authors.

As I have said in earlier rounds, this is a fascinating manuscript whose findings present a new perspective on a widely studied phenomenon. I do not agree with everything that is said, but the authors have sufficiently qualified their statements to the extent that the ideas are fully available for the reader to decide. I am sure this will inspire a great deal of future research.

Reviewer #3 (Remarks to the Author):

The authors have addressed the remaining issues. I have no further comments.